# NETWORK INSENSITIVITY TO PARAMETER NOISE VIA ADVERSARIAL REGULARIZATION

**Julian Büchel**
IBM Research - Zurich
SynSense, Zürich, Switzerland
ETH Zürich, Switzerland
jbu@zurich.ibm.com

**Fynn Faber**
ETH Zürich, Switzerland
faberf@ethz.ch

**Dylan R. Muir**
SynSense, Zürich, Switzerland
dylan.muir@synsense.ai

## ABSTRACT

Neuromorphic neural network processors, in the form of compute-in-memory cross-bar arrays of memristors, or in the form of subthreshold analog and mixed-signal ASICs, promise enormous advantages in compute density and energy efficiency for NN-based ML tasks. However, these technologies are prone to computational non-idealities, due to process variation and intrinsic device physics. This degrades the task performance of networks deployed to the processor, by introducing parameter noise into the deployed model. While it is possible to calibrate each device, or train networks individually for each processor, these approaches are expensive and impractical for commercial deployment. Alternative methods are therefore needed to train networks that are inherently robust against parameter variation, as a consequence of network architecture and parameters. We present a new network training algorithm that attacks network parameters during training, and promotes robust performance during inference in the face of random parameter variation. Our approach introduces a loss regularization term that penalizes the susceptibility of a network to weight perturbation. We compare against previous approaches for producing parameter insensitivity such as dropout, weight smoothing and introducing parameter noise during training. We show that our approach produces models that are more robust to random mismatch-induced parameter variation as well as to targeted parameter variation. Our approach finds minima in flatter locations in the weight-loss landscape compared with other approaches, highlighting that the networks found by our technique are less sensitive to parameter perturbation. Our work provides an approach to deploy neural network architectures to inference devices that suffer from computational non-idealities, with minimal loss of performance. This method will enable deployment at scale to novel energy-efficient computational substrates, promoting cheaper and more prevalent edge inference.

## 1 INTRODUCTION

There is increasing interest in NN and ML inference on IoT and embedded devices, which imposes energy constraints due to small battery capacity and untethered operation. Existing edge inference solutions based on CPUs or vector processing engines such as GPUs or TPUs are improving in energy efficiency, but still entail considerable energy cost (Huang et al., 2009). Alternative compute architectures such as memristor crossbar arrays and mixed-signal event-driven neural network accelerators promise significantly reduced energy consumption for edge inference tasks. Novel non-volatile memory technologies such as resistive RAM and phase-change materials (Chen, 2016; Yu & Chen, 2016) promise increased memory density with multiple bits per memory cell, as well as compact compute-in-memory for NN inference tasks (Sebastian et al., 2020). Analog implementations of neurons and synapses, coupled with asynchronous digital routing fabrics, permit high sparsity in both network architecture and activity, thereby reducing energy costs associated with computation.

However, both of these novel compute fabrics introduce complexity in the form of computational non-idealities, which do not exist for pure synchronous digital solutions. Some novel memory technologies support several bits per memory cell, but with uncertainty about the precise value stored on each cycle (Le Gallo et al., 2018b; Wu et al., 2019). Others exhibit significant drift in stored

states (Joshi et al., 2020). Inference processors based on analog and mixed-signal devices (Neckar et al., 2019; Moradi et al., 2018; Cassidy et al., 2016; Schemmel et al., 2010; Khaddam-Aljameh et al., 2022) exhibit parameter variation across the surface of a chip, and between chips, due to manufacturing process non-idealities. Collectively these processes known as "device mismatch" manifest as frozen parameter noise in weights and neuron parameters.

In all cases the mismatch between configured and implemented network parameters degrades the task performance by modifying the resulting mapping between input and output. Existing solutions for deploying networks to inference devices that exhibit mismatch mostly focus on per-device calibration or re-training (Ambrogio et al., 2018; Bauer et al., 2019; Nandakumar et al., 2020a). However, this, and other approaches such as few-shot learning or meta learning entail significant per-device handling costs, making them unfit for commercial deployment.

We consider a network to be "robust" if the output of a network to a given input does not change in the face of parameter perturbation. With this goal, network architectures that are intrinsically robust against device mismatch can be investigated (Thakur et al., 2018; Büchel et al., 2021). Another approach is to introduce parameter perturbations during training that promote robustness during inference, for example via random pruning (dropout) (Srivastava et al., 2014) or by injecting noise (Murray & Edwards, 1994).

In this paper we introduce a novel solution, by applying adversarial training approaches to parameter mismatch. Most existing adversarial training methods attack the input space. Here we describe an adversarial attack during training that seeks the parameter perturbation that causes the maximum degradation in network response. In summary, we make the following contributions:

- We propose a novel algorithm for gradient-based supervised training of networks that are robust against parameter mismatch, by performing adversarial training in the weight space.

- We demonstrate that our algorithm flattens the weight-loss landscape and therefore leads to models that are inherently more robust to parameter noise.

- We show that our approach outperforms existing methods in terms of robustness.

- We validate our algorithm on a highly accurate Phase Change Memory (PCM)-based Compute-in-Memory (CiM) simulator and achieve new state-of-the-art results in terms of performance and performance retention over time.

## 2 RELATED WORK

Research to date has focused mainly on adversarial attacks in the input space. With an increasing number of adversarial attacks, an increasing number of schemes defending against those attacks have been proposed (Wang et al., 2020; Zhang et al., 2019; Madry et al., 2019; Moosavi-Dezfooli et al., 2018). In contrast, adversarial attacks in parameter space have received little attention. Where parameter-space adversaries have been examined, it has been to enhance performance in semi-supervised learning (Cicek & Soatto, 2019), to improve robustness to *input-space* adversarial attacks (Wu et al., 2020), or to improve generalisation capability (Zheng et al., 2020).

We define "robustness" to mean that the network output should change only minimally in the face of a parameter perturbation — in other words, the weight-loss landscape should be as flat as possible at a loss minimum. Other algorithms that promote flat loss landscapes may therefore also be useful to promote robustness to parameter perturbations.

**Dropout** (Srivastava et al., 2014) is a widely used method to reduce overfitting. During training, a random subset of units are chosen with some probability, and these units are pruned from the network for a single trial or batch. This results in the network learning to distribute its computation across many units, and acts as a regularization against overfitting.

**Entropy-SGD** (Chaudhari et al., 2019) is a network optimisation method that minimises the local entropy around a solution in parameter space. This results in a smoothed parameter-loss landscape that should penalize sharp minima.

**Adversarial Block Coordinate Descent (ABCD)** (Cicek & Soatto, 2019) was proposed in order to complement input-space smoothing with weight-space smoothing in semi-supervised learning.

ABCD repeatedly picks half of the network weights and performs one step of gradient ascent on them, followed by applying gradient descent on the other half.

**Adversarial Weight Perturbation (AWP)** (Wu et al., 2020) was designed to improve the robustness of a network to adversarial attacks in the input space. The authors use Projected Gradient Ascent (PGA) on the network parameters to approximate a worst case perturbation of the weights $\Theta'$. PGA repeatedly computes the gradient of a loss function and updates the parameters in the direction of the (positive) gradient. After each update, the parameters are projected back onto a ball (e.g. in $l^2$) around the original parameters to ensure that a maximum distance is kept. Having identified an adversarial perturbation in the weight-space, an adversarial perturbation in the input-space is also found using PGA. Finally, the original weights $\Theta$ are updated using the gradient of the loss evaluated at the adversarial perturbation $\Theta'$.

**Adversarial Model Perturbation (AMP)** (Zheng et al., 2020) improves the generalisation of conventional neural networks by optimizing a standard loss evaluated using parameters that were perturbed adversarially using PGA. Unlike our method, (Zheng et al., 2020) did not formulate the loss function as a trade-off between performance and robustness. Furthermore, the presented algorithm, unlike our method, treats the perturbation $\Delta_\Theta$ to the parameters $\Theta$ as a constant during backpropagation.

**TRadeoff-inspired Adversarial DEfense via Surrogate-loss minimization (TRADES)** (Zhang et al., 2019) is a method for training networks that are robust against adversarial examples in the input space. The method consists of adding a boundary loss term to the loss function that measures how the network performance changes when the input is attacked. The boundary loss does not take the labels into account, so scaling it by a factor $\beta_{\text{rob}}$ allows for a principled trade-off between the robustness and the accuracy of the network.

**Noise injection during the forward pass** (Murray & Edwards, 1994) is a simple method for increasing network robustness to parameter noise. This method adds Gaussian noise to the network parameters during the forward pass and computes weight gradients with respect to the original parameters. This method regularizes the gradient magnitudes of output units with respect to the weights, thus enforcing distributed information processing and insensitivity to parameter noise. We refer to this method as "Forward Noise".

A recent paper proposed a method for improving the resilience to random and targeted bit errors in SRAM cells on digital Deep Neural Network (DNN) accelerators (Stutz et al., 2021). By employing adversarial or random bit flips during training, the authors significantly improved the robustness to bit perturbations, enabling the accelerators to be operated below the conventional supply voltage.

## 3 Methods

We use $\Theta$ to denote the set of parameters of a neural network $f(x, \Theta)$ that are trainable and susceptible to mismatch. The adversarial weights are denoted $\Theta^*$, where $\Theta_t^*$ are the adversarial weights at the $t$-th iteration of PGA. We denote the PGA-adversary as a function $\mathcal{A}$ that maps parameters $\Theta$ to attacking parameters $\Theta^*$. We denote a mini-batch of training examples as $X$ with $y$ being the corresponding ground-truth labels. $\prod_{\mathcal{E}_\zeta^p}(m)$ denotes the projection operator on the $\zeta$-ellipsoid in $l^p$ space. The operator $\odot$ denotes elementwise multiplication.

The effect of component mismatch on a network parameter can be modelled using a Gaussian distribution where the standard deviation depends on the parameter magnitude (Joshi et al., 2020; Büchel et al., 2021). In this paper we restrict ourselves to mismatch-driven perturbations in the network weights. For complex Spiking Neural Networks (SNNs), "network parameters" can refer to additional quantities such as neuronal and synaptic time constants or spiking thresholds. Our training approach described here can be equally applied to these additional parameters.

We define the value of an individual parameter when deployed on a neuromorphic chip as

$$\Theta^{\text{mismatch}} \sim \mathcal{N}(\Theta, \text{diag}(\zeta|\Theta|)) \tag{1}$$

where $\zeta$ governs the perturbation magnitude, referred to as the "mismatch level". The physics underlying the neuronal- and synaptic circuits lead to a model where the amount of noise introduced into the system depends linearly on the magnitude of the parameters. If mismatch-induced perturbations had constant standard deviation independent of weight values, one could use the weight-scale invariance

of neural networks as a means to achieve robustness, by simply scaling up all network weights (see Figure S4). The linear dependence of weight magnitude and mismatch noise precludes this approach.

In contrast to adversarial attacks in the *input space* (Carlini & Wagner, 2016; Moosavi-Dezfooli et al., 2015; Madry et al., 2019; Goodfellow et al., 2015), our method relies on adversarial attacks in *parameter space*. During training, we approximate the worst case perturbation of the network parameters using PGA and update the network parameters in order to mitigate these attacks. To trade-off robustness and performance, we use a surrogate loss (Zhang et al., 2019) to capture the difference in output between the normal and attacked network. Algorithm 1 illustrates the training procedure in more detail.

**begin**

  $\Theta_0^* \longleftarrow \Theta + |\Theta|\epsilon \odot R \; ; R \sim \mathcal{N}(0,1)$

  **for** $t = 1$ **to** $N_{steps}$ **do**

    $g \longleftarrow \nabla_{\Theta_{t-1}^*} \mathcal{L}_{\text{rob}}(\Theta, \Theta_{t-1}^*, X)$

    $v \longleftarrow \arg \max\limits_{v:\|v\|_p \leq 1} v^T g$

    $\Theta_t^* \leftarrow \prod_{\mathcal{E}_{\zeta_{\text{attack}}}^p} (\Theta_{t-1}^* + \alpha \odot v)$

  **end**

  $\Theta \longleftarrow \Theta - \eta \nabla_\Theta \mathcal{L}_{\text{nat}}((\Theta, X), y) + \beta_{\text{rob}} \mathcal{L}_{\text{rob}}(\Theta, \Theta_{N_{\text{steps}}}^*, X)$

**end**

**Algorithm 1:** In $l_\infty$, $v$ corresponds to $\text{sign}(g)$ and the step size $\alpha$ is $\frac{|\Theta| \odot \zeta}{N_{\text{steps}}}$. $\prod_{\mathcal{E}_{\zeta_{\text{attack}}}^p} (m)$ denotes the projection operator on the $\zeta_{\text{attack}}$-ellipsoid in $l^p$ space. In $l^\infty$ this corresponds to $\min(\max(m, \Theta - \epsilon), \Theta + \epsilon)$ with $\epsilon = \zeta_{\text{attack}} \odot |\Theta|$. $\zeta_{\text{attack}}$ and $\beta_{\text{rob}}$ are hyperparameters of our model.

Unlike adversarial training in the input space, where adversarial inputs can be seen as a form of data augmentation, adversarial training in the parameter space poses the following challenge: Because the parameters that are attacked are the same parameters being optimized, performing gradient descent using the same loss that was used for PGA would simply revert the previous updates and no learning would occur. ABCD circumvents this problem by masking one half of the parameters in the adversarial loop and masking the other half during the gradient descent step. However, this limits the adversary in its power, and requires multiple iterations to be performed in order to update all parameters at least once. AWP approached this problem by assuming that the gradient of the loss with respect to the attacking parameters can be used in order to update the original parameters to favor minima in flatter locations in weight-space. However, it is not clear whether this assumption always holds since the gradient of the loss with respect to the attacking parameters is not necessarily the same direction that would lead to a flatter region in the weight loss-landscape.

We approach this problem slightly differently: Similar to the TRADES algorithm (Zhang et al., 2019), our algorithm optimizes a natural (task) loss and a separate robustness loss.

$$\mathcal{L}_{\text{gen}}(\Theta, X, y) = \mathcal{L}_{\text{nat}}(\Theta, X, y) + \beta_{\text{rob}} \mathcal{L}_{\text{rob}}(\Theta, \mathcal{A}(\Theta), X)$$

Using a different loss for capturing the susceptibility of the network to adversarial attacks enables us to simultaneously optimise for performance and robustness, without PGA interfering with the gradient descent step. In our experiments, $\mathcal{L}_{\text{rob}}$ is defined as

$$\mathcal{L}_{\text{rob}}(\Theta, \Theta^*, X) = \text{KL}\left(f(\Theta, X), f(\Theta^*, X)\right) \tag{2}$$

This formulation comes with a large computational overhead since it requires computing the Jacobian $\mathbf{J}_{\Theta^*}(\Theta)$ of a complex recurrent relation between $\Theta$ and $\Theta^*$. To make our algorithm more efficient we assume that the Jacobian is diagonal, meaning that $\Theta^* = \Theta + \Delta\Theta$ for some $\Delta\Theta$ given by the adversary. In $l_\infty$, the Jacobian can then be calculated efficiently using (see suppl. material for details):

$$\mathbf{J}_{\Theta^*}(\Theta) = \mathbf{I} + \text{diag}\left[\frac{\text{sign}(\Theta) \odot (\zeta_{\text{attack}} + \epsilon \cdot R_1)}{N_{\text{steps}}} \odot \sum_{t=1}^{N_{\text{steps}}} \text{sign}\left(\nabla_{\Theta_t^*} \mathcal{L}_{\text{rob}}(\Theta, \Theta_t^*, X)\right)\right]$$

By making this assumption, our algorithm effectively multiplies the original training time by the number of PGA steps, similar to (Wu et al., 2020; Cicek & Soatto, 2019; Zheng et al., 2020).

Because component mismatch is independently proportional to the magnitude of each parameter, one has to model the space in which the adversary can search for a perturbation using an axis-aligned ellipsoid in $l_2$ and an axis-aligned box in $l_\infty$. Using an $\epsilon$-ball where the radius depends linearly on the individual parameter sets (Li et al., 2018; Cicek & Soatto, 2019; Wu et al., 2020) would either give the adversary too little or too much attack space. Projecting onto an axis-aligned ellipsoid in $l_2$ corresponds to solving the following optimization problem (Gabay & Mercier, 1976; Dai, 2006), which does not have a closed-form solution:

$$x^* = \arg \min_x \frac{1}{2} \|m - x\|_2$$
$$\text{s.t. } (x - c)^T W^{-2} (x - c) \leq 1$$

where $W = \text{diag}(|\Theta| \odot \zeta) + \mathbf{I} \cdot \zeta_{\text{const}}$, $c = \Theta$ and $m = \Theta^* + \alpha \odot v$. Because of the computational overhead this would incur, we only consider the $l_\infty$ case in our experiments.

## 4 RESULTS

The ultra-low power consumption of mixed-signal neuromorphic chips make them suitable for edge-applications, such as always-on voice detection (Cho et al., 2019), vibration monitoring (Gies et al., 2021) or always-on face recognition (Liu et al., 2019). For this reason, we consider two compact network architectures in our experiments: A Long Short-term spiking recurrent Neural Network (LSNN) with roughly 65k trainable parameters; a conventional CNN with roughly 500k trainable parameters; and a Resnet32 architecture (He et al., 2015) (see Supplementary Material S1 for more information). We trained models to perform four different tasks:

- Speech command detection of 6 classes (Warden, 2018);
- ECG-anomaly detection on 4 classes (Bauer et al., 2019);
- Fashion-MNIST (F-MNIST): clothing-image classification on 10 classes (Xiao et al., 2017); and
- The Cifar10 colour image classification task (Krizhevsky, 2009).

We compared several training and attack methods, beginning with a standard Stochastic Gradient Descent (SGD) approach using the Adam optimizer (Kingma & Ba, 2015) ("Standard"). Learning rate varied by architecture, but was kept constant when comparing training methods on an architecture. We examined networks trained with dropout (Srivastava et al., 2014), AWP (Wu et al., 2020), AMP (Zheng et al., 2020), ABCD (Cicek & Soatto, 2019), and Entropy-SGD (Chaudhari et al., 2019). The adversarial perturbations used in AWP and ABCD were adapted to our mismatch model (i.e. magnitude-dependent in $l_\infty$) unless stated otherwise. AMP was not adapted.

A dropout probability of 0.3 was used in the dropout models and $\gamma$ in AWP was set to 0.1. When Gaussian noise was applied to the weights during the forward pass (Murray & Edwards, 1994) a relative standard deviation of 0.3 times the weight magnitude was used ($\eta_{\text{train}} = 0.3$). For Entropy-SGD, we set the number of inner iterations to 10 with a Langevin learning rate of 0.1. Because Entropy-SGD and ABCD have inner loops, the number of total epochs were reduced accordingly. All other models were trained for the same number of epochs (no early stopping) and the model with the highest validation accuracy was selected.

**Effectiveness of adversarial weight attack** We examined the strength of our adversarial weight attack during inference and training. Standard networks trained using gradient descent alone with no additional regularization (Fig. S5a, "Standard") were disrupted badly by our adversarial attack during inference ($\zeta = 0.1$; final mean test accuracy $91.40\% \rightarrow 17.50\%$), and this was not ameliorated by further training. When our adversarial attack was implemented during training (Fig. S5a, $\beta_{\text{rob}} = 0.1$), the trained network was protected from disruption both during training and during inference (final test accuracy $91.97\% \rightarrow 78.41\%$).

Our adversarial attack degrades network performance significantly more than a random perturbation. Because our adversary uses PGA during the attack, it approximates a worst-case perturbation of the network within an ellipsoid around the nominal weights $\Theta$. We compared the effect of our attack against a random weight perturbation (random point on $\zeta$−ellipsoid) of equal magnitude.

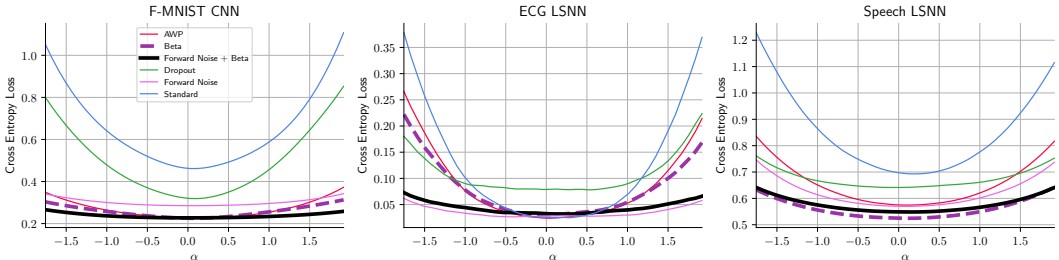

Figure 1: **Our training method flattens the test weight-loss landscape.** When moving away from the trained weight minimum ($\alpha = 0$) in randomly-chosen directions, we find that our adversarial training method (Beta; Beta+Forward) finds deeper minima (for F-MNIST and Speech tasks) at flatter locations in the cross-entropy test loss landscape. See text for further details, and Fig. S2 for visualisation over several random seeds.

For increasing perturbation size during inference (Fig. S5c; $\zeta$), our adversarial attack disrupted the performance of the standard network significantly more than a random perturbation (test acc. $91.40\% \rightarrow 17.50\%$ (attack) vs. $91.40\% \rightarrow 90.63\%$ (random) for $\zeta = 0.1$). When our adversarial attack was applied during training (Fig. S5d; $\beta_{\text{rob}} = 0.1$), the network was protected against both random and adversarial attacks for magnitudes up to $\zeta = 0.7$ and $\zeta = 0.1$, respectively.

**Flatness of the weight-loss landscape** Under our definition of robustness, the network output should change only minimally when the network parameters are perturbed. This corresponds to a loss surface that is close to flat in weight space. We measured the test weight-loss landscape for trained networks, compared over alternative training methods and for several architectures (Fig. 1). We examined only cross-entropy loss over the test set, and not the adversarial attack loss component (KL divergence loss; see Eq. 2). For each trained network, we chose a random vector $v \sim \mathcal{N}(\mathbf{0}, \zeta|\Theta|)$ and calculated $\mathcal{L}_{\text{cce}}(f(X_{\text{test}}, \Theta + \alpha \cdot v), y_{\text{test}})$ for many evenly-spaced $\alpha \in [-2, 2]$. This process was repeated 5 times for $\zeta = 0.2$, and the means plotted in Fig. 1. Weight-loss landscapes for the individual trials are shown in Fig. S2.

Our adversarial training approach found minima of trained parameters $\Theta$ in flatter areas of the weight-loss landscape, compared with all other approaches examined (flatter curves in Fig. 1). In most cases our training approach also found deeper minima at lower categorical cross-entropy loss ($\mathcal{L}_{\text{cce}}$), reflecting better task performance. These results are reflected in the better generalization performance of our approach (see Table 1). Not surprisingly, dropout and AWP also lead to flatter minima than the Standard network with no regularization. ABCD and Entropy-SGD were not included in Fig. 1 because they did not outperform the Standard model.

**Network robustness against parameter mismatch** We evaluated the ability of our training method to protect against simulated device mismatch. We introduced frozen parameter noise into models trained with adversarial attack, with noise modelled on that observed in neuromorphic processors (Joshi et al., 2020; Büchel et al., 2021). In these devices, uncertainty associated with each weight parameter is approximately normally distributed around the nominal value, with a standard deviation that scales with the weight magnitude (Eq 1). We measured test accuracy under simulated random mismatch for 100 samples across two model instances. A comparison of our method against standard training is shown in Fig. 2. For mismatch levels up to 70% ($\zeta = 0.7$), our approach protected significantly against simulated mismatch for all three tasks examined ($p < 2 \times 10^{-8}$ in all cases; U test). A more detailed comparison between the different models is given in Table 1.

**Network robustness against direct adversarial attack on task performance** Our training approach improves network robustness against mismatch parameter noise. We further evaluated the robustness of our trained networks against a parameter adversary that directly attacks task performance, by performing PGA on the cross-entropy loss $\mathcal{L}_{cce}$. Note that this is separate from the adversary used in our training method, which attacks the boundary loss (Eq.2). The AWP method uses the cross-entropy loss to find adversarial parameters during training. Nevertheless, we found that

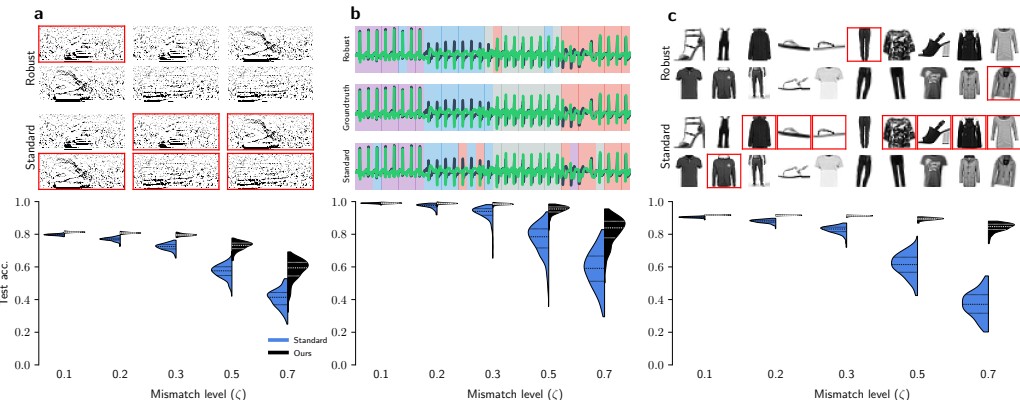

Figure 2: **Adversarial attack during training protects networks against random mismatch-induced parameter noise.** Networks were evaluated for the Speech (a), ECG (b) and F-MNIST tasks (c), under increasing levels of simulated mismatch ($\zeta$). Networks trained using standard SGD were disrupted by mismatch levels $\zeta > 0.1$. At all mismatch levels our adversarial training approach performed significantly better in the presence of mismatch (higher test accuracy). Red boxes highlight misclassified examples.

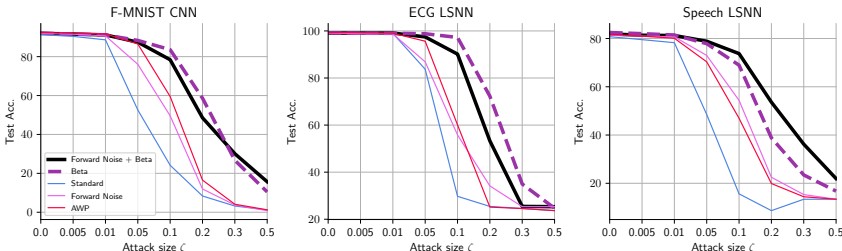

Figure 3: **Our training method protects against task-adversarial attacks in parameter space.** Networks trained under several methods were attacked using a PGA adversary that directly attacked the task performance $\mathcal{L}_{cce}$. Our adversarial training approach (bold; dashed) outperformed all other methods against this attack.

our method consistently outperforms all other compared methods for increasing attack magnitude $\zeta$ (Fig. 3). In networks trained with our method, the adversary needed to perform a considerably larger attack to significantly reduce performance (test accuracy < 70%). ABCD and Entropy-SGD were not included in the comparison because they did not outperform the standard network.

**Robustness against parameter drift for PCM based CiM** CiM devices based on memristor technologies such as PCM promise to deliver energy- and space-efficient accelerators. With the increasing interest in CiM devices for accelerated inference and energy-efficient edge computing (Sebastian et al., 2020), the problem of deploying a model that is robust to noise originating from the device physics of PCM cells has gained in significance. We investigated the effect of our training method on the robustness of networks that are simulated to run on PCM-based CiM hardware (for details on the simulator, see SM 5).

Networks trained with our method outperform state-of-the-art networks deployed on PCM-based CiM. Currently, the method that has proven to yield the best performance on CiM hardware is training with noise on the network parameters during the forward pass. We adapted this method by adding our algorithm and show that we consistently outperform the conventional method (see Figure S14) for a wide range of hyperparameters, and even surpass the FP baseline (a model trained without noise injection and evaluated on a standard PC) for some configurations (see Figure 4). Following

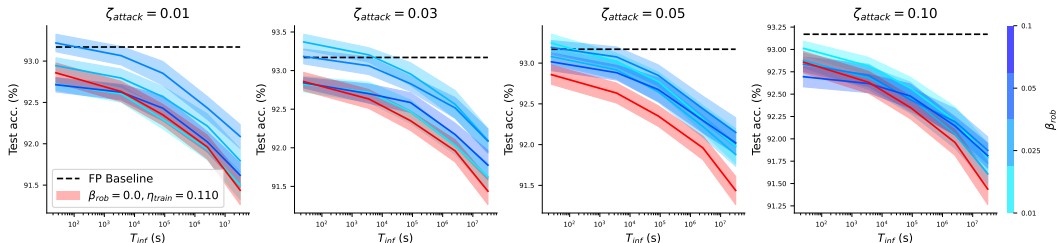

Figure 4: **Networks trained with our method show overall better performance when deployed on PCM-based CiM hardware.** This figure shows the performance degradation as a consequence of the PCM devices drifting over time (x-axis, up to one year) of networks deployed on CiM hardware. Each subplot shows networks trained with a different attacking magnitude ($\zeta_{\text{attack}}$) that are trained with different values of $\beta_{\text{rob}}$. Each network is compared to the FP-baseline and a network trained with Gaussian noise injection on the weights.

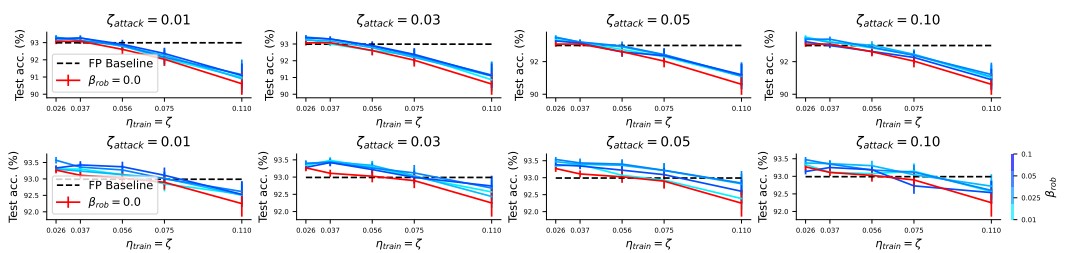

Figure 5: **Our method consistently yields networks that outperform training with Gaussian noise injection.** This figure compares the robustness to Gaussian noise at various levels ($\zeta$) for networks trained with our method (blue) and a networks trained with Gaussian noise injection (red), where the level of noise used during training ($\eta_{\text{train}}$) matches the noise used during inference. Each row represents a different type of noise: The first row models Gaussian noise with a standard deviation that is proportional to the largest absolute weight in the individual weight kernels (Joshi et al., 2020) and the second row follows the model presented in this paper (see Eq. 1).

experiments conducted in (Joshi et al., 2020), we used Resnet32 (He et al., 2015) trained on Cifar10 (Krizhevsky, 2009).

Injecting Gaussian noise on the weights during the forward pass yields strong improvements compared to the standard network. We show that by adding our method, we consistently improve this robustness by a significant amount (see Figure 5).

We furthermore improve the scalability of our method by using a pretrained model and fewer steps for the adversary. Our algorithm incurs an additional training time that scales linearly with the number of attack steps used in the adversary (note that we cache the necessary gradients for the Jacobian calculation). To alleviate this additional time, we show that our method produces good results even for just one single adversarial step. Figure S13 shows the resulting performance when varying the number of attack steps used by the adversary. It should be noted that all results reported on PCM robustness were obtained using three adversarial steps and a pretrained model.

**Verifiable robustness for LSNNs** We investigated the provable robustness of LSNNs trained using our method using abstract interpretation (Cousot & Cousot, 1977; Gehr et al., 2018; Mirman et al., 2018). In this analysis a function $f(x, \Theta)$ (in our case, a neural network with input $x$ and parameters $\Theta$) is overapproximated using an *abstract domain*. We specify the weights $\Theta$ in our network as an interval parameterised by the attack size $\zeta$, spanning $[\Theta - \zeta|\Theta|, \Theta + \zeta|\Theta|]$. We examined the proportion of provably correctly classified test samples for the Speech and ECG tasks, under a range of attack sizes $\zeta$, and comparing our approach against standard gradient descent and against training

with forward-pass noise only (Fig. S9). We found that our approach is provably more correct over increasing attack size $\zeta$ (higher verified test accuracy).

## 5 Discussion

We proposed a new training approach that includes adversarial attacks on the parameter space during training. Our proposed adversarial attack was significantly stronger than random weight perturbations at disrupting the performance of a trained network. Including the adversarial attack during training significantly protected the trained network from weight perturbations during inference. Our approach found minima in the weight-loss landscape that usually corresponded to lower loss values, and were always in flatter regions of the loss landscape. This indicates that our approach found network solutions that are less sensitive to parameter variation, and therefore more robust. Our approach was more robust than several other methods for inducing robustness and good generalisation. To the best of our knowledge, our work represents the first example of interval bound propagation applied to SNNs, and the first application of parameter-space adversarial attacks to promote network robustness against device mismatch for mixed-signal compute. Our experiments only considered the impact of weight perturbations, and did not examine the influence of uncertainty in other parameters of mixed-signal neuromorphic processors such as time constants or spiking thresholds. Our approach can be adapted to include adversarial attacks in the full network parameter space, increasing the robustness of spiking networks. The technique of interval bound propagation can also be applied to these additional network parameters. We did not quantize network parameters either during or after training, in this work. On some platforms (Moradi et al., 2018) it is necessary to deploy quantized weights and it is unclear how our adversarial attacks would interact with quantization during the training process. However, most PCM-based CiM hardware does not require quantization during training in order to get good performance (Joshi et al., 2020). Per-device training for a device with known calibrated parameter noise is likely to achieve the highest possible deployed performance on that single device. However, this approach has significant drawbacks. Firstly, each device must be either measured / calibrated accurately — not a trivial requirement — or trained with the device in the forward inference pass of the training loop. Secondly, training must be performed individually for each device, entailing significant logistical problems if the training is conducted in the factory or inside a consumer product. Thirdly, this approach will retain full sensitivity to parameter variation on the device. Our method improves the performance of neural networks deployed to inference hardware that include computational non-idealities. For example, NN processors with crossbar architectures based on novel memory devices such as RRAM and PCM (Sebastian et al., 2020) display uncertainty in stored memory values as well as conductance drift over time (Le Gallo et al., 2018b; Wu et al., 2019; Joshi et al., 2020). Our method could also address in-memory computing-based NN processors based on SRAM and switched capacitors (Verma et al., 2019). Analog neurons and synapses in mixed-signal NN processors, for example SNN inference processors (Moradi et al., 2018), exhibit variation in weights and neuron parameters across a processor. We showed that our training approach finds network solutions that are insensitive to mismatch-induced parameter variation. Our networks can therefore be deployed to inference devices with computational non-idealities with only minimal reduction in task performance, and without requiring per-device calibration or model training. This reduction in per-device handling implies a considerable reduction in expense when deploying at commercial scale. Our method therefore brings low-power neuromorphic inference processors closer to commercial viability.

**Ethics statement** The authors declare no conflicts of interest.

**Reproducibility statement** Code for reproduce all experiments described in this work are provided at https://github.com/jubueche/BPTT-Lipschitzness and https://github.com/jubueche/Resnet32-ICLR

**Acknowledgments** This work was partially supported by EU grants 826655 "TEMPO"; 871371 "MEMSCALES"; and 876925 "ANDANTE" to DRM. JB would also like to thank Manuel Le Gallo-Bourdeau, Irem Boybat and Abu Sebastian from IBM Research - Zurich, for insightful discussions and technical support.

Table 1: **Results of training multiple networks using several methods over three tasks.** Networks were evaluated under different levels of mismatch ($\zeta$).

**CNN**

| Mismatch | Forward Noise, $\beta_{rob} = 0.1$ | | | $\beta_{rob} = 0.25$ | | | Standard | | | Forward Noise | | | AWP ($\epsilon_{pga} = 0.0$) | | | Dropout | | | AMP ($\epsilon = 0.005$) | | |
|---|---|---|---|---|---|---|---|---|---|---|---|---|---|---|---|---|---|---|---|---|---|
| | Mean Acc. | Std. | Min. | Mean Acc. | Std. | Min. | Mean Acc. | Std. | Min. | Mean Acc. | Std. | Min. | Mean Acc. | Std. | Min. | Mean Acc. | Std. | Min. | Mean Acc. | Std. | Min. |
| Baseline (0.0) | 91.88 | **0.00** | 91.88 | 92.11 | 0.22 | 91.89 | 91.30 | 0.15 | 91.15 | 92.00 | 0.02 | 91.98 | 92.35 | 0.12 | **92.23** | **92.42** | 0.24 | 92.18 | 91.34 | 0.12 | 91.22 |
| 0.1 | 91.77 | 0.12 | 91.29 | 91.59 | 0.26 | 90.91 | 90.45 | 0.37 | 88.80 | 91.94 | **0.09** | 91.69 | **92.19** | 0.16 | **91.78** | 91.13 | 0.85 | 86.50 | 90.42 | 0.35 | 89.37 |
| 0.2 | 91.62 | **0.16** | **91.19** | 90.67 | 0.42 | 89.28 | 87.93 | 1.03 | 83.40 | **91.63** | **0.16** | 91.06 | 91.42 | 0.44 | 89.87 | 88.71 | 1.55 | 80.97 | 87.97 | 0.88 | 85.33 |
| 0.3 | **91.25** | **0.22** | **90.36** | 89.64 | 0.65 | 87.10 | 82.70 | 2.45 | 71.99 | 91.01 | 0.26 | 90.01 | 89.88 | 0.95 | 85.11 | 84.94 | 3.06 | 72.97 | 83.13 | 2.13 | 75.73 |
| 0.5 | **89.36** | **0.73** | **86.84** | 85.96 | 1.87 | 76.28 | 61.14 | 6.92 | 42.46 | 87.82 | 0.88 | 84.37 | 82.59 | 3.66 | 66.73 | 71.74 | 5.84 | 53.10 | 60.60 | 7.36 | 38.67 |
| 0.7 | **84.19** | 2.63 | **74.25** | 79.39 | 4.40 | 59.15 | 36.93 | 7.73 | 20.17 | 78.48 | 2.95 | 69.67 | 65.38 | 8.27 | 40.66 | 53.91 | 9.03 | 22.49 | 36.79 | 7.42 | 18.53 |

**ECG LSNN**

| Mismatch | Forward Noise, $\beta_{rob} = 0.1$ | | | $\beta_{rob} = 0.25$ | | | Standard | | | Forward Noise | | | AWP ($\epsilon_{pga} = 0.0$) | | | Dropout | | | AMP ($\epsilon = 0.02$) | | |
|---|---|---|---|---|---|---|---|---|---|---|---|---|---|---|---|---|---|---|---|---|---|
| | Mean Acc. | Std. | Min. | Mean Acc. | Std. | Min. | Mean Acc. | Std. | Min. | Mean Acc. | Std. | Min. | Mean Acc. | Std. | Min. | Mean Acc. | Std. | Min. | Mean Acc. | Std. | Min. |
| Baseline (0.0) | 99.07 | 0.34 | 98.73 | 99.10 | 0.07 | 99.03 | 99.07 | **0.04** | **99.03** | 99.25 | 0.37 | 98.88 | 99.22 | 0.19 | 99.03 | 98.10 | 0.11 | 97.99 | **99.40** | **0.00** | **99.40** |
| 0.1 | 99.04 | 0.27 | 98.36 | 98.96 | 0.24 | 98.28 | 98.95 | 0.27 | 97.76 | **99.09** | 0.19 | **98.66** | 98.93 | 0.31 | 97.69 | 97.71 | 0.34 | 96.72 | 99.04 | 0.38 | 97.39 |
| 0.2 | 98.87 | 0.35 | 96.94 | 98.16 | 0.71 | 93.96 | 97.22 | 1.46 | 91.87 | **99.01** | 0.26 | **98.06** | 97.71 | 0.97 | 93.06 | 96.89 | 0.87 | 93.81 | 97.03 | 1.52 | 90.67 |
| 0.3 | 98.45 | **0.45** | **96.49** | 96.34 | 1.95 | 89.33 | 92.97 | 4.38 | 65.30 | **98.59** | 0.52 | 95.30 | 94.60 | 2.93 | 81.34 | 94.85 | 2.47 | 85.60 | 92.24 | 3.92 | 76.27 |
| 0.5 | **94.86** | **2.69** | 82.39 | 86.22 | 6.66 | 60.75 | 76.55 | 9.61 | 35.75 | 94.44 | 2.86 | **82.84** | 80.32 | 8.08 | 41.94 | 87.56 | 6.05 | 64.10 | 73.36 | 11.27 | 29.55 |
| 0.7 | **82.02** | 8.40 | **50.22** | 70.07 | 11.16 | 39.33 | 58.67 | 11.44 | 29.48 | 80.00 | 8.70 | 37.69 | 63.44 | 9.61 | 32.91 | 74.24 | 12.58 | 30.15 | 56.64 | 11.02 | 27.76 |

**Speech LSNN**

| Mismatch | Forward Noise, $\beta_{rob} = 0.5$ | | | $\beta_{rob} = 0.5$ | | | Standard | | | Forward Noise | | | AWP ($\epsilon_{pga} = 0.01$) | | | Dropout | | | AMP ($\epsilon = 0.01$) | | |
|---|---|---|---|---|---|---|---|---|---|---|---|---|---|---|---|---|---|---|---|---|---|
| | Mean Acc. | Std. | Min. | Mean Acc. | Std. | Min. | Mean Acc. | Std. | Min. | Mean Acc. | Std. | Min. | Mean Acc. | Std. | Min. | Mean Acc. | Std. | Min. | Mean Acc. | Std. | Min. |
| Baseline (0.0) | 81.52 | 0.05 | 81.47 | **82.48** | **0.00** | **82.48** | 80.86 | 0.03 | 80.83 | 81.33 | 0.14 | 81.20 | 82.38 | 0.14 | 82.25 | 79.02 | 0.36 | 78.66 | 80.83 | 0.44 | 80.39 |
| 0.1 | 81.33 | **0.24** | 80.72 | 82.01 | 0.31 | **81.03** | 79.72 | 0.45 | 78.53 | 81.12 | 0.32 | 80.18 | **82.03** | 0.37 | 80.79 | 79.16 | 0.44 | 77.88 | 80.00 | 0.55 | 78.49 |
| 0.2 | 80.77 | **0.35** | 79.71 | **81.24** | 0.43 | **79.98** | 76.96 | 1.11 | 72.57 | 80.10 | 0.53 | 78.73 | 80.58 | 0.65 | 78.69 | 78.58 | 0.67 | 75.85 | 77.56 | 0.87 | 74.81 |
| 0.3 | 79.57 | **0.62** | **77.98** | **79.80** | 0.67 | 77.54 | 72.31 | 2.00 | 65.40 | 78.05 | 0.88 | 75.45 | 77.47 | 1.35 | 70.88 | 77.18 | 0.96 | 74.43 | 73.80 | 1.61 | 68.24 |
| 0.5 | 72.67 | 2.75 | **63.85** | **73.40** | 2.32 | 60.91 | 57.16 | 4.25 | 42.07 | 67.41 | 3.93 | 53.50 | 63.98 | 4.28 | 47.38 | 69.80 | 3.13 | 54.75 | 59.91 | 4.49 | 44.74 |
| 0.7 | 58.03 | 6.57 | 32.19 | **60.23** | 4.66 | **45.49** | 40.70 | 5.72 | 24.92 | 49.19 | 6.98 | 30.47 | 44.83 | 6.65 | 23.13 | 55.96 | 5.74 | 37.74 | 42.88 | 6.83 | 25.03 |

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

## SUPPLEMENTARY MATERIAL

### SPIKING RNN ARCHITECTURE

The dynamics of the spiking model (Bellec et al., 2018) can be summarised by a set of differential equations (Eq. 3).

$$
\begin{aligned}
\boldsymbol{B}^t &= \boldsymbol{b}^0 + \beta \boldsymbol{b}^t \\
\boldsymbol{o}^t &= \mathbf{1}(\boldsymbol{V}^t > \boldsymbol{B}^t) \text{ unless refractory } t_{\text{refr}} \\
\boldsymbol{b}^{t+1} &= \boldsymbol{\rho}_\beta \boldsymbol{b}^t + (1 - \boldsymbol{\rho}_\beta)\frac{\boldsymbol{o}^t}{\text{dt}} \\
\boldsymbol{I}^t_{\text{Reset}} &= \frac{\boldsymbol{o}^t}{\text{dt}} \boldsymbol{B}^t \text{dt} \\
\boldsymbol{V}^{t+1} &= \boldsymbol{\rho}_V \boldsymbol{V}^t + (1 - \boldsymbol{\rho}_V)(\boldsymbol{I}_{\text{in}} W_{\text{in}} + \frac{\boldsymbol{o}^t}{\text{dt}} W_{\text{rec}}) - \boldsymbol{I}^t_{\text{Reset}}
\end{aligned}
\tag{3}
$$

where $\boldsymbol{\rho}_\beta = \mathrm{e}^{-\text{dt}/\tau_{\text{ada}}}$ and $\boldsymbol{\rho}_V = \mathrm{e}^{-\text{dt}/\tau}$. The variables $\boldsymbol{B}$ describe the spiking thresholds with spike-frequency adaptation. The vector $\boldsymbol{o}^t$ denotes the population spike train at time $t$. The membrane potentials $\boldsymbol{V}$ have a time constant $\tau$ and the adaptive threshold time constant is denoted by $\tau_{\text{ada}}$. The speech signals and ECG traces fed into the network are represented as currents $\boldsymbol{I}_{\text{in}}$. Since the derivative of the spiking function with respect to its input is mostly 0 we use a surrogate gradient that is explicitly defined as

$$
\frac{\partial E}{\partial \boldsymbol{V}^t} = \frac{\partial E}{\partial \boldsymbol{z}^t}\frac{\partial \boldsymbol{z}^t}{\partial \boldsymbol{V}^t} = \frac{\partial E}{\partial \boldsymbol{z}^t} d \cdot \max(\mathbf{1} - |\frac{\boldsymbol{V}^t - \boldsymbol{B}^t}{\boldsymbol{B}^t}|, \mathbf{0})
\tag{4}
$$

where $E$ is the error and $d$ is the dampening factor. To get the final prediction of the network, we average the population spike trains along the time-axis $\boldsymbol{z}_{\text{avg}}$ and compute

$$
\begin{aligned}
\boldsymbol{l} &= \text{softmax}(\boldsymbol{z}_{\text{avg}} W_{\text{out}} + b_{\text{out}}) \\
\hat{y} &= \arg\max_i \boldsymbol{l}_i
\end{aligned}
$$

### CNN ARCHITECTURE

Our architecture comprises two convolutional blocks ($2 \times [4 \times 4, 64$ channels, MaxPool, ReLU]), followed by three dense layers ($N = 1600, 256, 64$, ReLU) and a softmax layer. All weights and kernels are initialized using the Glorot normal initialisation (Glorot & Bengio, 2010). Using this architecture, we achieved a test accuracy of $\sim 93\%$. Attacked parameters for this network included all the kernel weights, as well as all the dense layer parameters.

DERIVATION OF JACOBIAN

Under the assumption that $\alpha = \frac{\zeta_{\text{attack}} \odot |\Theta|}{N_{\text{steps}}}$ and $p = \infty$, we can rewrite the inner loop of Algorithm 1 to

**begin**
$\quad \Theta^* \longleftarrow \Theta + |\Theta| \epsilon \odot R \ ; R \sim \mathcal{N}(0, 1)$
$\quad$ **for** $t = 1$ **to** $N_{\text{steps}}$ **do**
$\quad\quad \Theta_t^* \leftarrow \Theta_{t-1}^* + \alpha \cdot \text{sign}\left(\nabla_{\Theta_{t-1}^*} \mathcal{L}_{\text{rob}}(\Theta, \Theta_{t-1}^*, X)\right)$
$\quad$ **end**
**end**

By rewriting $\Theta^*$ in the form of $\Theta^* = \Theta + \Delta\Theta$ we get

$$\Theta^* = \Theta + |\Theta|\epsilon \odot R + \alpha \odot \sum_{t=1}^{N_{\text{steps}}} \text{sign}\left(\nabla_{\Theta_{t-1}^*} \mathcal{L}_{\text{rob}}(f(\Theta, X), f(\Theta_{t-1}^*, X))\right)$$

From this the Jacobian can be easily calculated. Plugging in the defintion for $\alpha$ we get

$$\mathbf{J}_{\Theta^*}(\Theta) = \mathbf{I} + \text{diag}\left[\frac{\text{sign}(\Theta) \odot (\zeta_{\text{attack}} + \epsilon \cdot R_1)}{N_{\text{steps}}} \odot \sum_{t=1}^{N_{\text{steps}}} \text{sign}\left(\nabla_{\Theta_t^*} \mathcal{L}_{\text{rob}}(f(\Theta, X), f(\Theta_t^*, X))\right)\right]$$

MISMATCH MODEL

To model the parameter noise introduced by component mismatch we used a Gaussian distribution where the mean is the nominal noise-free weight value and the standard deviation depends linearly on the weight value. This model realistically captures the behaviour of parameter mismatch on a mixed-signal neuromorphic SNN inference processor (Moradi et al., 2018). Fig. S1 shows the quantified parameter mismatch recorded directly from neuromorphic HW, over a range of nominal parameter values and for several neuronal and synaptic parameters. The measured mismatch parameter variation follows an approximately Gaussian distribution where the standard deviation depends linearly on the mean.

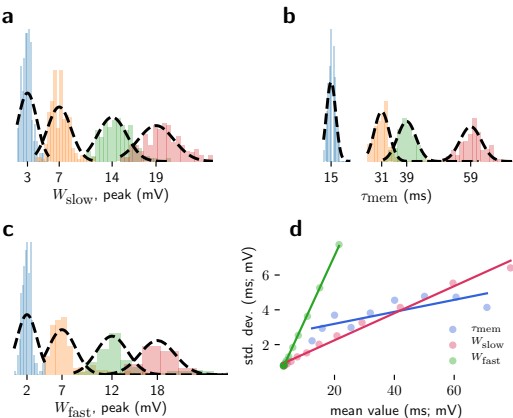

Figure S1: **Quantification of mismatch on analog neuromorphic hardware.** Parameter values for several weight parameters and membrane time constants were measured for a range of nominal parameter values, using an oscilloscope directly connected to a mixed-signal neuromorphic SNN processor. (**a-c**) Various parameters of the chip follow a Gaussian distribution with increasing width. (**d**) The mismatch standard deviation depends linearly on the nominal value of each parameter.

WEIGHT LOSS-LANDSCAPE VISUALIZATION

We characterized the shape of the weight loss-landscape by plotting the categorical cross entropy loss for varying levels of noise added to the weights of the trained network. In each trial, we picked a random vector $v \sim \mathcal{N}(\mathbf{0}, \zeta |\Theta|)$ and evaluated the categorical cross entropy of the whole test set given the weights $\Theta + \alpha \cdot v$, where $\alpha \in [-2, 2]$ and $\zeta = 0.2$. As we show in Figure S2, the variance of the individual 1D weight loss-landscapes is small.

Table S1 quantifies the flatness of the illustrated weight loss-landscapes. As can be seen, our method combined with adding noise during the forward pass yields the flattest landscapes.

Table S1: **Average slope of estimated 1D weight loss-landscapes.** Slopes were calculated as the mean absolute differences between sample points divided by the sampling distance.

|  | F-MNIST CNN | ECG LSNN | Speech LSNN |
|---|---|---|---|
| Standard | 0.3375 | 0.1605 | 0.2702 |
| Beta | 0.0480 | 0.0879 | 0.0633 |
| Forward Noise | 0.0332 | **0.0180** | 0.1009 |
| Forward Noise + Beta | **0.0190** | 0.0187 | **0.0540** |
| Dropout | 0.3022 | 0.0952 | 0.0707 |
| AWP | 0.0841 | 0.1013 | 0.1389 |

Figure S2: **Illustration of the test weight loss-landscape highlighting individual trials to measure the loss landscapes.** See the results text for more details.

ATTACKING KL-DIVERGENCE LOSS DURING INFERENCE

While the adversary in AWP attacks the task loss directly, i.e. max $\mathcal{L}_{\text{cce}}(f(\Theta^*, X), y)$, the adversary in our training algorithm attacks the KL divergence max $\text{KL}(f(\Theta, X), f(\Theta^*, X))$. This implies that our parameter attack seeks simply to change the response of the network in any way, and is agnostic to the task itself. In the main results we attack the cross-entropy task loss during inference, in order to not give an undue advantage to our training approach. Here we show that our training approach also provides robustness against parameter attacks on the KL-divergence loss during inference.

Fig. S3 shows the adversarial robustness of the methods for an adversary that attacks the KL-divergence rather than the cross-entropy loss. When the network is attacked by maximizing the KL-divergence between the normal and attacked network, our adversarially-trained networks are more robust than AWP, SGD and forward-noise. Because the parameter attack used here during inference as well as in the inner optimization loop of the training procedure is the same, this result is expected, and serves as a sanity check that our networks indeed learn to defend against the attack they were trained against.

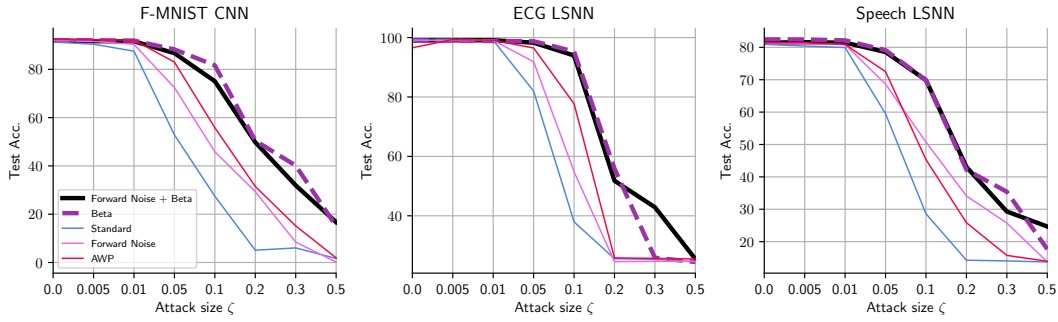

Figure S3: **Robustness to weight attack targeting KL divergence during inference.** When the network is attacked by maximizing the KL-divergence between the normal and attacked network, our adversarially trained networks are more robust than standard SGD or AWP.

EFFECT OF CONSTANT VERSUS RELATIVE PARAMETER NOISE

As described in the main text, parameter noise that has constant magnitude (for example, Gaussian noise with fixed standard deviation) is trivial to protect against by increasing weight magnitudes. We examined this effect by training MLPs with an adversary that employs Gaussian noise with fixed std. dev. $\epsilon = 0.2$. Figure S4 (right) illustrates the test accuracy of two MLPs trained on F-MNIST over the course of training. Using an inverse weight decay term, the weight-magnitude of one network is forced to increase to 2.0 over the course of training (black crosses, $\Theta^*$). The weight magnitude of the other network (red crosses) is limited during training to 0.2 (red crosses; $\Theta$). One can observe that as the weight magnitude of the increasing magnitude network $\Theta^*$ increases, also the robustness to Gaussian noise ($\epsilon = 0.2$) increases (blue), while the performance of the small magnitude network $\Theta$ remains poor (cyan).

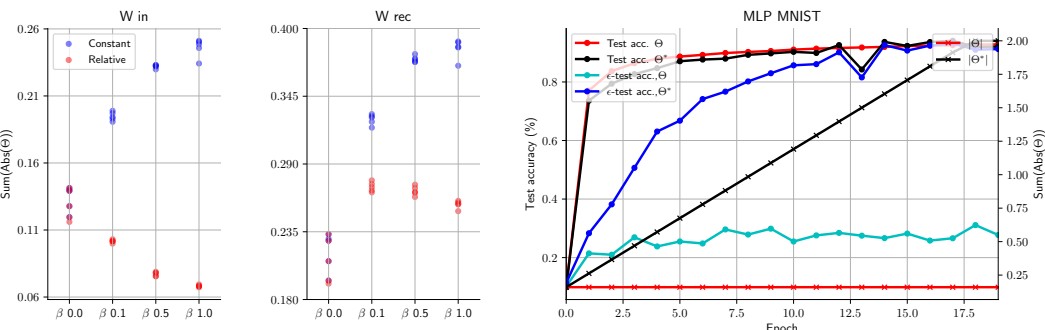

Figure S4: **Constant magnitude versus magnitude-relative parameter noise.** (left, middle) When parameter noise of constant magnitude (blue) is introduced by the adversarial attack during training, the networks learn to increase the magnitude of the weights to trivially improve robustness to the constant-magnitude attack. When the parameter attack is relative to each parameter magnitude, as in the main text (red), the weight magnitudes do not increase. These networks were trained for a range of $\beta_{\text{rob}}$, i.e. varying emphasis on robustness.
(right) One can also trivially increase the robustness to fixed-magnitude noise by introducing an inverse weight decay term that causes the weights to increase in magnitude (black, cross marker) while retaining performance. This causes the MLP trained on MNIST to become increasingly robust to fixed-magnitude parameter noise (blue dots; $\epsilon$-test acc., $\Theta^*$). The network where the weight magnitude was not increased over time (red, cross marker) did not improve in terms of robustness (cyan; $\epsilon$-test acc., $\Theta$), although both models perform similarly when there is no parameter noise applied (red and black dots).

EFFECT OF ADVERSARIAL REGULARIZATION DURING TRAINING

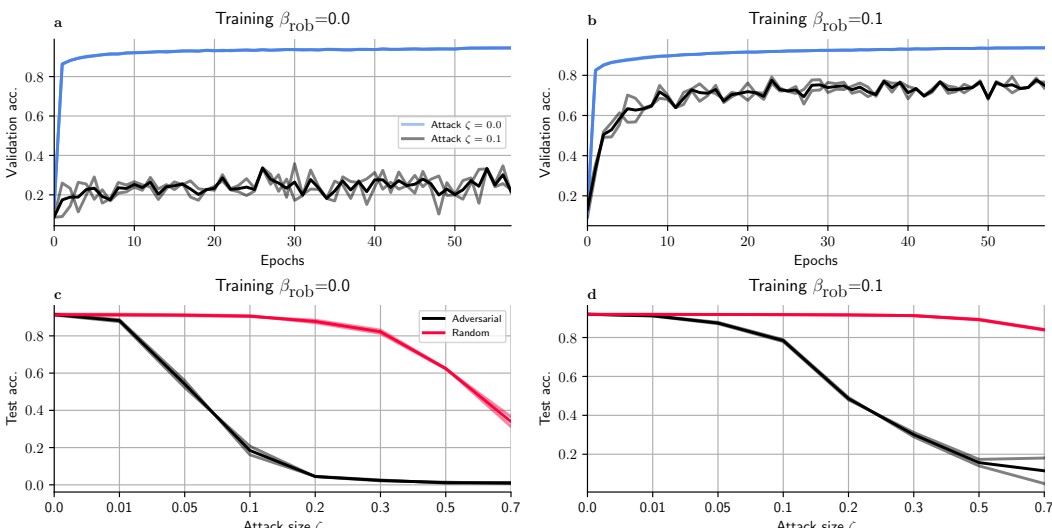

Figure S5: **Our parameter attack is effective at decreasing the performance of a network during inference, and using our attack during training protects a network from later disruption.** (a) A network trained using standard SGD (i.e. $\beta_{\mathrm{rob}} = 0.0$) on the F-MNIST task is disrupted badly by the parameter noise adversary during inference ($\zeta = 0.1$; black curve). (b) When the same network is trained with parameter attacks during training ($\beta_{\mathrm{rob}} = 0.1$), the network is protected from parameter attacks during inference (high accuracy of attacked network; black curve). (c) When trained with standard SGD ($\beta_{\mathrm{rob}} = 0.0$), both random noise (red) and parameter attacks (black) disrupt network performance for increasing attack size $\zeta$. (d) Under our training approach ($\beta_{\mathrm{rob}} = 0.1$), networks are significantly protected against random and adversarial weight perturbations during inference.

EFFECT OF VARYING $\beta_{\text{rob}}$

We additionally quantify the trade-off between test loss and robustness of our algorithm by repeating the experiment in Figure 1 at different values of $\beta_{\text{rob}}$. As shown in Figure S6, increasing $\beta_{\text{rob}}$ flattens the weight loss-landscape, effectively increasing the robustness of the model. To further substantiate this claim, we repeated the experiment of Figure S5 with the same values of $\beta_{\text{rob}}$. As Figure S8 shows, increasing $\beta_{\text{rob}}$ and therefore increasing the flatness of the landscape yields increased robustness to random, as well as, adversarial perturbations.

We note that, relative to the baseline, the test loss does not consistently increase with increasing values of $\beta_{\text{rob}}$. We hypothesize that this is due to the increased generalization capability of our networks. To check whether this is indeed the case, we repeated the experiment from Fig. S6 using the loss computed on the training set. As Fig. S7 shows, increasing values of $\beta_{\text{rob}}$ lead to flatter minima and higher loss on the training set, which is the exact trade-off to be expected from the formulation of our loss function. However, the increased generalization capability that follows from a flatter loss-landscape seems to disrupt this trade-off. As a result, when choosing $\beta_{\text{rob}}$ one should aim at choosing the highest value that still yields good performance on the validation set.

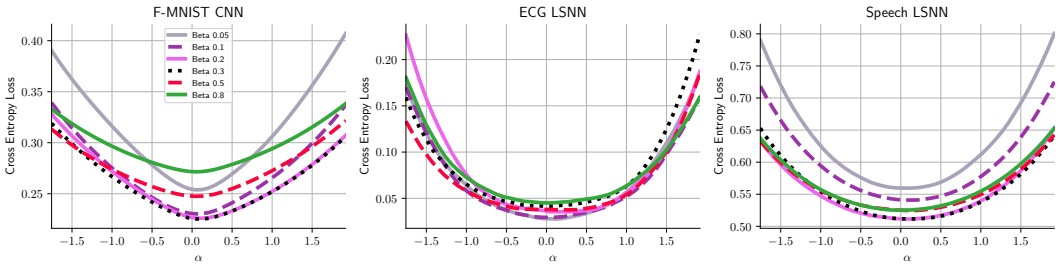

Figure S6: **Effect of $\beta_{\text{rob}}$ on the test loss landscape.** Increasing $\beta_{\text{rob}}$ promotes robustness by flattening the test loss landscape. However, increasing $\beta_{\text{rob}}$ does not lead to a systematic rise in loss on the test set.

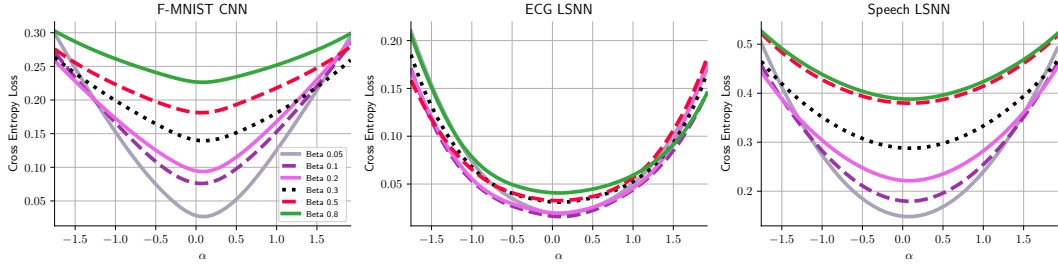

Figure S7: **Effect of $\beta_{\text{rob}}$ on the training loss landscape.** Weight loss landscape computed on the training set (c.f. Fig. S6). Increasing $\beta_{\text{rob}}$ leads to flatter loss-landscapes (increased robustness) as before, but also a systematic increase in training loss (higher values for cross-entropy loss; note ordering of curves).

We additionally trained the CNN on the F-MNIST data for a range of $\beta_{\mathrm{rob}}$, and measured the network robustness to attacks of varying magnitude $\zeta$ (Fig. S8). We compared the effectiveness of random attack versus adversarial attack during inference, evaluated on the test set. We found that increasing $\beta_{\mathrm{rob}}$ improved the robustness of the network to both random and adversarial attack during inference.

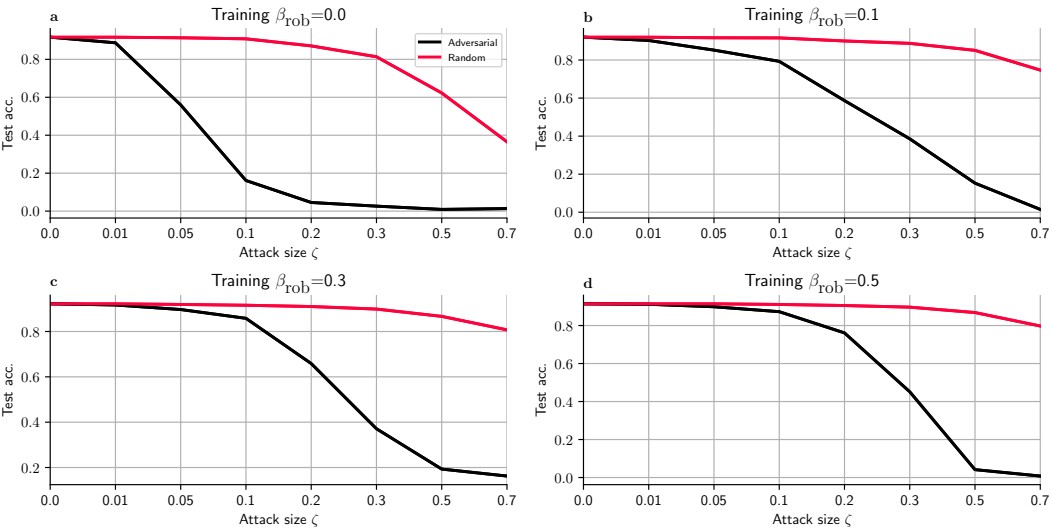

Figure S8: **Increasing $\beta_{\mathrm{rob}}$ during training improves robustness during inference.** The CNN was trained using different values of $\beta_{\mathrm{rob}}$ and evaluated on the test set after the weights were perturbed either randomly (red) or adversarially (black).

VERIFIABLE ROBUSTNESS FOR LSNNS

Computations through the network are performed on intervals rather than on discrete values. By propagating the intervals through a network over a test set, we obtain output logits that are also expressed as intervals and can therefore determine whether a sample will always be classified correctly.

The final classification of the network is made using an $\arg\max$ operator. For provability of network performance, we consider that a test sample $x$ is correctly classified when the lower bound of the logit interval for the correct class is the maximum lower bound across all logit intervals, and when the logit interval for the correct class is disjoint from the other logit intervals. When the logit interval for the correct class overlaps with another logit interval, that test sample is not considered to be provably correctly classified. Note that interval domain analysis provides a relatively loose bound (Gehr et al., 2018), with the implication that the results here probably underestimate the true performance of our method.

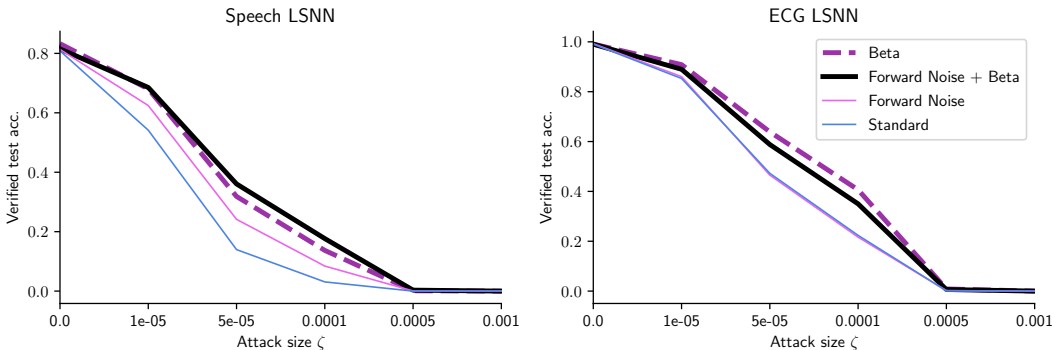

Figure S9: **Networks trained with our method are provably more robust than those trained with standard gradient descent or forward noise alone.** We used interval bound propagation to determine the proportion of test samples that are verifiably correctly classified, under increasing weight perturbations $\zeta$. For both the Speech and ECG tasks, computed on the trained LSNNs, our method was provably more robust for $\zeta < 5 \times 10^{-4}$.

WIDE-MARGIN NETWORK ACTIVATIONS

Murray et al. show that adding random forward noise to the weights of a network during the forward pass implicitly adds a regularizer of the form $\Theta_{i,j}^2 \left( \frac{\partial o_{k,l}}{\partial \Theta_{i,j}} \right)^2$ to the network weights (Murray & Edwards, 1994). When sigmoid activation functions are used, this regularizer favors high or low activations. When implementing interval bound propagation for LSNNs, intervals over spiking activity must be computed by passing intervals through the spiking threshold function. As a result, intervals for spiking activity become either $[0, 0]$ for neurons that never emit a spike regardless of weight attack; $[1, 1]$ for neurons that always emit a spike; and $[0, 1]$ for neurons for which activity becomes uncertain in the presence of weight attack. By definition, a robust network should promote bounds $[0, 0]$ and $[1, 1]$, where the activity of the network is unchanged by weight attack. Robust network configurations should therefore avoid states where the membrane potentials of neurons are close to the firing threshold.

To see whether this was also the case for our LSNNs, we investigated the distribution of the membrane potentials on a batch of test examples for a network that was trained with- and without noise during the forward pass. We found that robust networks exhibited a broader distribution of membrane potentials, with comparatively less distribution mass close to the firing threshold (Fig. S10). This indicates that neurons in the robust network spend more time in a "safe" regime where a small change in the weights cannot trigger an unwanted spike, or remove a desired spike.

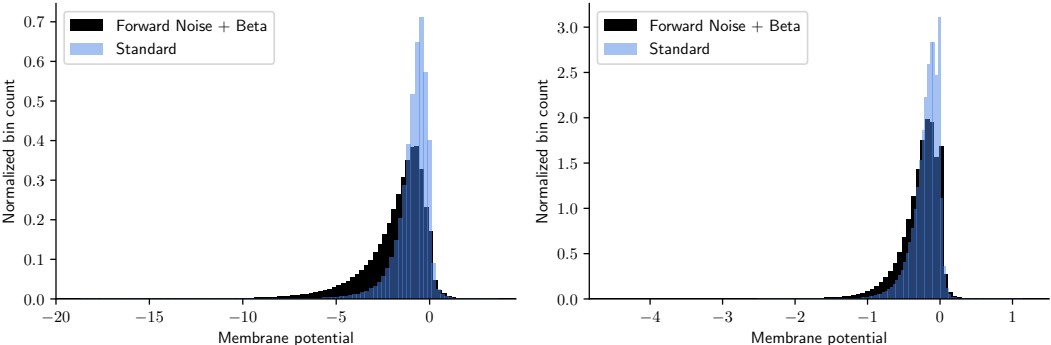

Figure S10: **Membrane potentials are distributed away from the firing threshold for networks trained with forward noise.** We measured the distribution of membrane potentials in LSNNs trained using standard gradient descent ("Standard"), and in the presence of weight noise injected in the forward pass ("Forward Noise + Beta"). As predicted, networks trained with forward noise have membrane potentials distributed away from the firing threshold. This implies that weight perturbations are less likely to inject or delete a spike erroneously, improving the robustness of the network.

EFFECT OF VARYING $\epsilon_{\text{PGA}}$ IN AWP

In addition to attacking the network parameters, AWP (Wu et al., 2020) also attacks the input using PGA. Since the relation between robustness to weight- and input-space perturbations is still unclear, we performed additional sweeps over the attack size in the input space. Figure S11 demonstrates that attacking the input during training generally does not improve the robustness, with the exception for the network trained on the speech dataset. We also note that attacking the inputs improved robustness for larger mismatch values, but generally degraded performance for the small values.

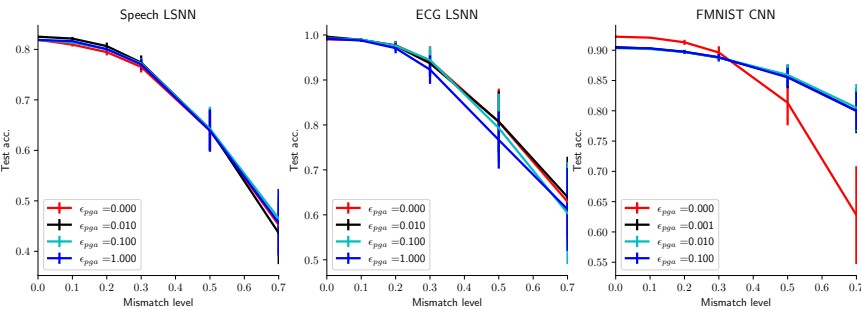

Figure S11: **Attacking the input using conventional PGA generally has a limited effect on robustness to weight-space perturbations.** We swept various values of $\epsilon_{\text{pga}}$, the parameter that determines the maximum perturbation in $l^\infty$ for the AWP algorithm.

TRAINING OF CNN USED FOR PCM-BASED CiM SIMULATION

The CNN that was used for this series of experiments is Resnet32 (He et al., 2015) trained on the Cifar10 (Krizhevsky, 2009) dataset. The CNN was generally trained for 300 epochs using a batch size of 256. We used SGD with an initial learning rate of 0.001 that was decreased by a multiplicative factor of 0.2 after epochs 60, 120 and 160. Additionally, Nesterov momentum (Nesterov, 1983) was used with a value of 0.9 and weight decay with a value of 5e-4.

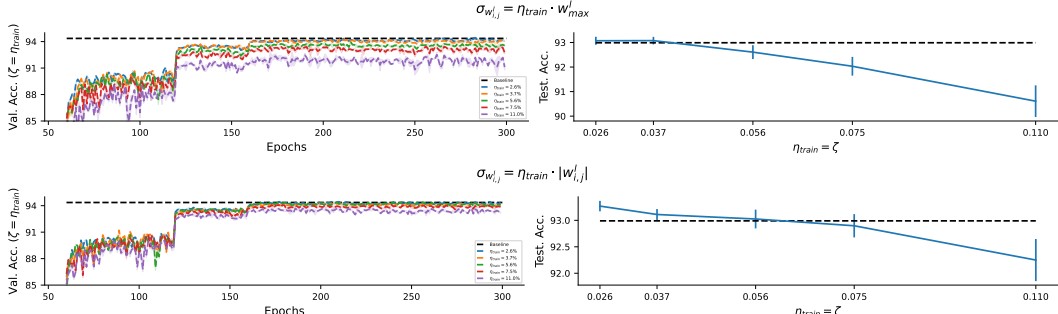

Figure S12: The left panel illustrates the validation accuracy over the course of finetuning a pretrained model for additional 240 epochs. One should note that convergence is usually achieved much quicker (roughly after 150 additional epochs). The right panel illustrates the performance on the test set of each model trained with noise injection of magnitude $\eta_{\text{train}}$ (x-axis). Each row depicts a different noise model. It should be noted that the weights in each filter were clipped to two standard deviations during training to avoid outliers causing excessive amounts of noise following the model that relies on the maximum weight value.

VARYING THE NUMBER OF ATTACK STEPS FOR THE PCM-BASED CiM SIMULATION

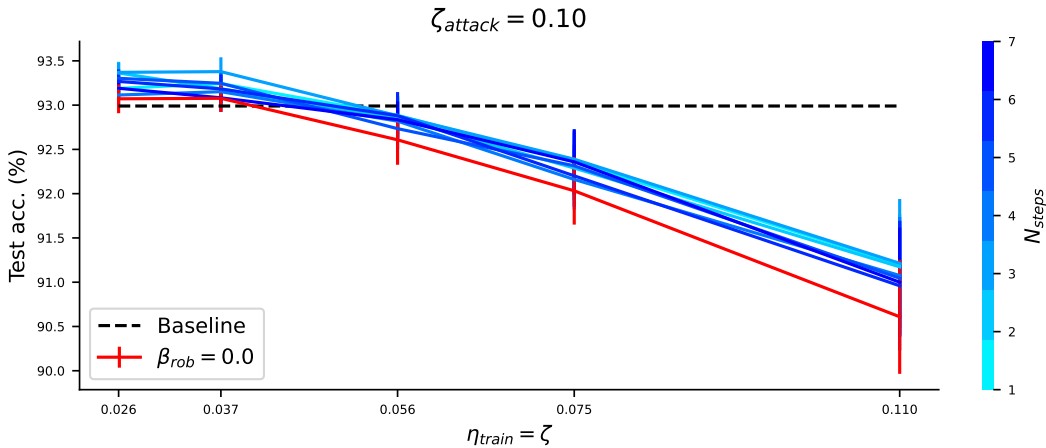

Figure S13: **One can greatly reduce the number of attack steps used during training.** Our method still produces strong results for very few number of attack steps (blue) when compared to the baseline model (red, trained with Gaussian noise ($\eta_{\text{train}}$)).

PERFORMANCE OF VARYING HYPERPARAMETERS FOR THE PCM-BASED CIM SIMULATION

In this experiment we show that the choice of hyperparameters is generally not very important to outperform the baseline that was trained with noise injection. However, in order to surpass the FP baseline, i.e. the model that was trained without noise injection and evaluated on a standard PC, one has to tune the hyperparameters in order to obtain a combination that yields the highest performance. Figure S14 illustrates this sweep. Each row represents a different value of $\eta_{\text{train}}$ that was used for training the baseline model (red). Each column represents a different attack size and the different hues of blue correspond to varying values of $\beta_{\text{rob}}$.

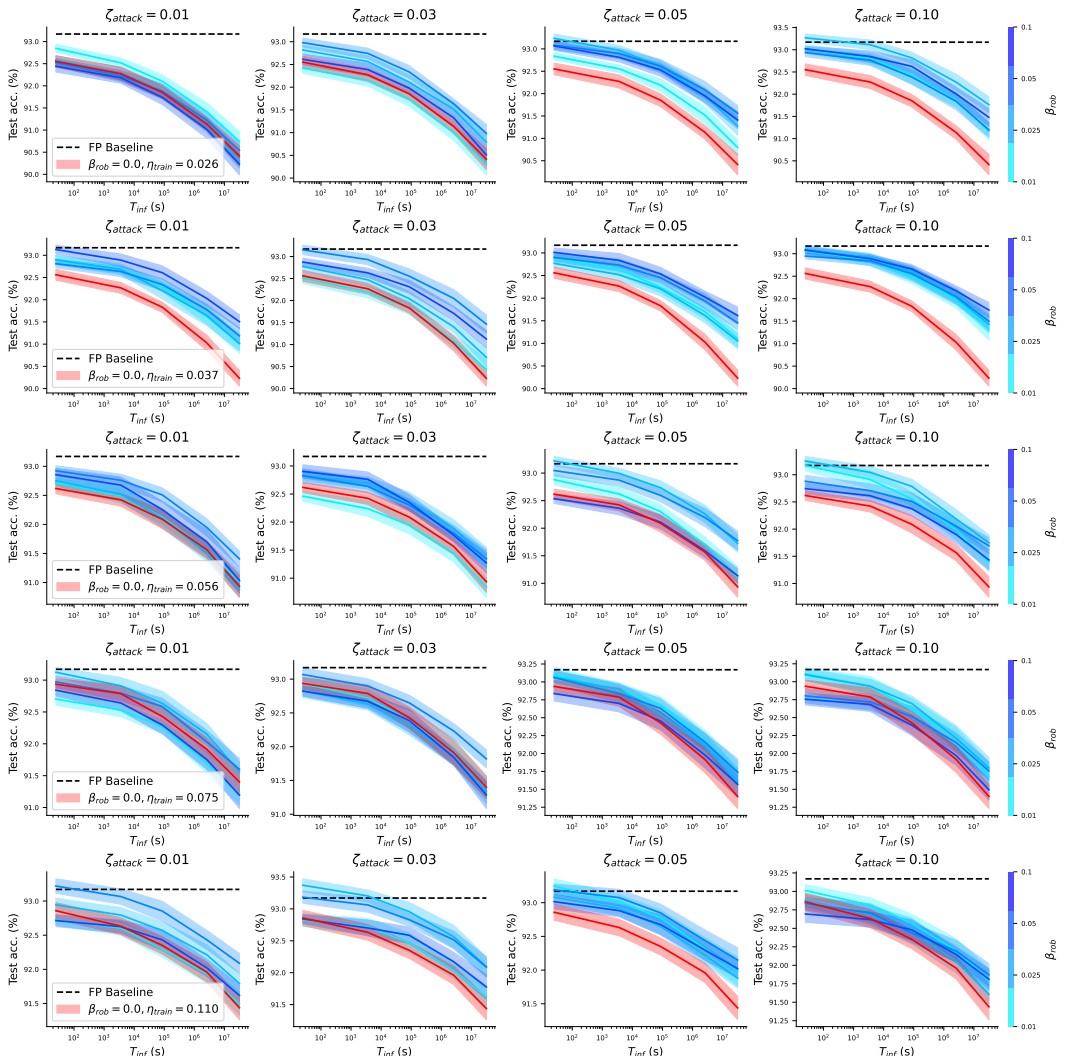

Figure S14: **The choice of hyperparameters is not critical in order to beat the baseline model.** Our method is resilient to variations in hyperparameters. However, to obtain configurations where even the FP baseline is surpassed, one has to fine-tune the method.

PCM NOISE MODEL

Analog CiM comes in various flavors, depending on the memory technology used. In this paper, we assume the use of PCM devices, which have been heavily studied in the context of analog CiM accelerators (Joshi et al., 2020; Nandakumar et al., 2019; Boybat et al., 2018). PCM-based, or, more generally, Non-Volatile Memory (NVM)-based architectures, essentially perform Matrix-Vector-Multiplications (MVMs) using Kirchhoff's current law. The weights of the matrix are organized as differential pairs in order to account for positive and negative weights. When storing a neural network, each weight matrix is programmed into the NVM devices by applying short electrical pulses (Nandakumar et al., 2020b). Because of various noise sources, this process is often imprecise and exhibits noise on the weights, termed "programming noise". Additionally, PCM devices suffer from $1/f$ and telegraph noise, adding even more noise during inference ("read noise"). At last, PCM devices also drift due to the underlying physical properties (Le Gallo et al., 2018a). Although the effect of drift can mostly be alleviated by scaling the output of the MVM (a method called Global Drift Compensation (GDC)), the non-uniform drift of the devices still leads to performance degradation over time. In the simulator that we used, we model these three main sources of noise, analog-to-digital and digital-to-analog converters, GDC and splitting of the MVM to account for smaller tile sizes (typically each crossbar is $256 \times 256$).

Initially, the clipped weights are mapped to target conductances in a differential manner, i.e. the weight matrix is split into two conductance matrices that both represent the positive and negative weights as conductances. The target conductances typically range from zero to $G_{\max}$, where $G_{\max}$ is assumed to be $25\mu S$. After mapping the weights to the target conductances $G_T$, the programming noise is simulated (these statistical models assume the conductances to be normalized): $G_P = G_T + \mathcal{N}(0, \sigma_P)$ where $\sigma_P = \max(-1.1731G_T^2 + 1.9650G_T + 0.2635, 0.0)$.

After the conductances have been programmed they drift over time, with the conductance of a device typically following $G_D = G_P(t/t_c)^{-\nu}$, where $\nu$ is the drift coefficient, $t$ is the time at inference, and $t_c$ is the time the conductances were programmed. Additionally, the drift coefficient is modelled to follow a Gaussian distribution. This makes it typically hard to correct for drift and it is the main reason why drift is a problem in PCM based CiM devices.

Finally, the read noise is modelled using a Gaussian: $G_R \sim \mathcal{N}(G_D, \sigma_{nG(t)})$, where $\sigma_{nG(t)} = G_D(t)Q\sqrt{log((t + t_r)/t_r)}$ with $Q = \min(0.0088/G_T^{0.65}, 0.2)$ and $t_r = 250ns$ and $t$ is the time at inference. For the experiments in this paper, we simulated the performance of the networks deployed on CiM hardware for up to one year.

