# OpenReview forum: "NETWORK INSENSITIVITY TO PARAMETER NOISE VIA PARAMETER ATTACK DURING TRAINING"
_ICLR.cc/2022/Conference — ICLR 2022 Poster_

### Official Review · Reviewer_vgst · 2021-11-02

**Correctness:** 4
**Technical Novelty And Significance:** 3
**Empirical Novelty And Significance:** 3
**Recommendation:** 6
**Confidence:** 2

**Main Review:**

First of all, this paper is well-written and easy to follow.
The authors provided clear explanations on the prior works, and the proposed method is clearly described.
The proposed method that uses the robustness loss seems like a simple idea, but the experimental results showed clear improvement compared to existing methods with several different benchmarks.

However, the datasets used for experiments seem too small in terms of the network size and the number of parameters, so it is doubtful that the proposed method can be a scalable solution.
Since there seems no reason for the proposed method to be limited to the spiking neural networks, it would be better if the authors can showed the improvement from the proposed method with larger-scale datasets commonly used for non-spiking neural networks.


**Summary Of The Paper:**

In this paper, the authors proposed an adversarial training method that minimizes the robustness loss so that the network parameter can be trained to be robust to the parameter perturbation.
The motivation of the work comes from the parameter mismatch that exists when using analog devices for computation.
The robustness loss calculates the difference between the network output before the weight perturbation and after the weight perturbation, and the direction of perturbation is decided in adversarial manner with Projected Gradient Ascent (PGA).
Experimental results with several benchmarks showed that minimizing such loss resulted in flattened weight-loss landscape, and the proposed method outperformed existing methods in terms of the robustness to parameter mismatch.

**Summary Of The Review:**

This paper proposed a simple yet effective solution for parameter mismatch with gradient-based adversarial training method.
However, since I am not familiar with the prior works on the parameter mismatch, I am not very confident about my assessment.

---

> ### Author Response · Authors · 2021-11-18
> **We address scalability and perform additional experiments on standard ML & in-memory computing benchmarks**
>
> _[...] the datasets used for experiments seem too small in terms of the network size and the number of parameters, so it is doubtful that the proposed method can be a scalable solution_
>
> - The additional training time that is incurred by our algorithm scales linearly with the number of attack steps used by the adversary. Because one can cache the gradients necessary for computing the Jacobian (which is just a diagonal matrix), the only bottleneck of our algorithm is the gradient calculation done by PGA in the adversary. To alleviate this effect, we conducted additional experiments in which we show that very few attack steps typically suffice to get good performance. We show that one can even use just one attack step in combination with a pre-trained model to outperform the runner-up method.
>
> _[...] it would be better if the authors can showed the improvement from the proposed method with larger-scale datasets commonly used for non-spiking neural networks._
>
> - The reviewer is correct that this method is not only applicable to spiking networks and we have conducted additional experiments using a more standard architecture (Resnet32) and dataset (Cifar10) that is not only common in ML research, but also in in-memory computing research [1,2,3].
>
> [1] A Programmable Neural-Network Inference Accelerator Based on Scalable In-Memory Computing, (Jia et al.), _ISSCC 2021_
>
> [2] Accurate deep neural network inference using computational phase-change memory, (Joshi et al.), _Nature communications_
>
> [3] Mitigating Imperfections in Mixed-Signal Neuromorphic Circuits, (Fahimi et al.), _Preprint_

---

> > ### Comment · Reviewer_vgst · 2021-11-23
> > **I think there is still a limitation in the network/problem size being dealt with**
> >
> > Thanks for the comments and really sorry for answering back late.
> > My concern on the network/problem size simply came from the fact that Resnet32 and CIFAR-10, which are the biggest (in terms of parameter/input size) network/problem used in this paper, are still small compared to those used in many other papers being published in ML conferences.
> > As far as I know, those relatively small network/problems are still being used in several research subtopics, including the Spiking Neural Networks or hardware-based ML research, due to their limitations in time-efficient training/inference.
> >
> > I think the idea proposed in this paper is not limited to such special cases, and there seems to be no justification for conducting experiments using only smaller network/problems. Especially considering the fact that ICLR is a ML conference and not a hardware conference.

---

> > > ### Author Response · Authors · 2021-11-24
> > > **Benchmarks**
> > >
> > > We thank the reviewer for their feedback. We included a larger architecture with a larger scale problem, in response to your concerns.
> > >
> > > Our paper is a methods paper, and we demonstrate the applicability and benefit of our method to several widely ranging tasks over several widely ranging architectures, including two cutting-edge computational substrates (memristors and SNNs).
> > >
> > > There is no reason to think that our method would behave differently or worse on a larger benchmark dataset — and measuring performance on any specific benchmark is not really the aim of our ML methods paper. We hope that you can understand our perspective here.

---

### Official Review · Reviewer_9GjF · 2021-11-04

**Correctness:** 4
**Technical Novelty And Significance:** 3
**Empirical Novelty And Significance:** 3
**Recommendation:** 8
**Confidence:** 3

**Main Review:**

This paper presents a novel application of a loss formulation (task loss and robustness loss) and projected gradient descent training to build resilience to parameter noise into neural networks.  The paper is presented well and provides sufficient evidence on the performance of the method.  While the grammar and layout is generally good, Figure caption text size does need to be increased in some cases (e.g. Fig 1 and 2 legends).  It appears details are present to reproduce results from a casual glance at supplementary code in conjunction with details provided in the paper.   Repeated and averaged results in several figures strengthens confidence of reported figures.cce

I am unsure of the implications of assuming the Jacobian of the parameters w.r.t. the attack parameters is diagonal but I believe this would limit the device applications to those where noise is uncorrelated between parameters.  Correlated noise may also have a small effect on final network performance even when trained under a diagonal Jacobian assumption.

L_{cce} is referred to twice in the paper but is undefined -- is this the same as L_{nat} task loss?



**Summary Of The Paper:**

This paper introduces a special loss formulation and training algorithm leveraging projected gradient ascent to build robustness to parameter noise into training for neural networks (both spiking and convolutional neural networks are tested).  The authors show that the proposed method flattens the loss landscape for small parameter perturbations with a tradeoff of increased overall cross entropy (task) loss.  It is shown that this alternative loss, when utilized during training, provides stronger resiliency to adversarially injected parameter noise than competing methods such as dropout or adversarial weight perturbation.  Tests relating to device mismatch due to manufacturing process deficiencies are also performed and the authors show their training method would be useful for building networks that don't require per-device tuning in such cases (neuromorphic/analog computing).

**Summary Of The Review:**

Techniques in this paper appear to be novel modifications to neural network training that combine projected gradient ascent with a loss specifically designed to promote robustness to small perturbations in network weights (or any similarly learned parameter).  The authors provide evidence to the strengths of this technique when compared with other mitigation measures for noise in the parameter space (either adversarial or as a byproduct of physical properties).  This research provides an interesting path toward making imperfect analog computing devices practical for large scale use and deployment.  Though I am not an expert in the specifics of this adversarial and noise related network training methods, the paper layout and methodology appear sound.

---

> ### Author Response · Authors · 2021-11-18
> **We investigate drift in Phase-Change-Memory (PCM) devices used in in-memory computing**
>
> _[...] Correlated noise may also have a small effect on final network performance even when trained under a diagonal Jacobian assumption._
>
>
> - Many devices experience correlated noise when used under different temperatures or when the devices drift over time. As we show in a new set of experiments, our method also increases the robustness to drift prominent in phase-change-memory devices. For Resnet32 trained on Cifar10, we were able to reduce the performance degradation over one year to just 1%. We, however, believe that it would be an interesting topic for future research to investigate methods for mitigating correlated noise in more detail.
>
> $L_{cce}$ _is referred to twice in the paper but is undefined -- is this the same as_ $L_{nat}$ _task loss?_
>
> - We thank the reviewer for pointing out that $L_{cce}$ is not defined properly. This is fixed in the revised version. In general, the two losses are not the same, as the natural loss can be any loss you choose (e.g. categorical cross entropy with an additional regularisation term). In our experiments, these two were actually the same.

---

### Official Review · Reviewer_BXMW · 2021-11-04

**Correctness:** 4
**Technical Novelty And Significance:** 2
**Empirical Novelty And Significance:** 3
**Recommendation:** 6
**Confidence:** 4

**Main Review:**

The paper is clear to understand in most parts. Experiments are well presented, including successful demonstrations of better flattening of the weight-loss landscape with respect to the competitive methods. My main concern relates to the ambiguity on explicitly putting the novelty of this paper as the proposed problem is recently addressed by similar studies. In fact, adversarial parameter perturbations for regularization is an approach that was demonstrated in the context of model generalization with similar results (see below). The novelty of this paper however, in my opinion, comes from the fact that the authors applied this idea for the first time to recurrent spiking neural networks (SNNs).

Given the current narrative, title, and manuscript organization, the methodological contribution of this work is not directly put to the domain of compact SNN architectures and utility in the context of neuromorphic hardware, however sort of generally implied. As CNN experiments are also demonstrated to show that the proposed idea extends to conventional networks, one obvious concern would be on the scalability of this approach on the parameter space to larger networks as training would get costly to estimate Eq (2). So far the conventional CNN that the authors used only has 500k trainable parameters, and is not even demonstrated to operate on RGB images such as CIFAR-10. Did the authors investigate this with respect to the competitive methods?

From various aspects the work has similarities to the adversarial model perturbation (AMP) framework that was recently proposed [Zheng et al., Regularizing Neural Networks via Adversarial Model Perturbation, CVPR 2021]. A very similar goal was reached to obtain robust parameters to variations, including convergence to flatter local minima regions for empirical risk minimization. While I agree on the choice of all the other competitive methods that the authors compared their approach with in the paper, one needs to compare/discuss AMP with the proposed method (which uses a novel TRADES-like loss objective).

It is ambiguous what does the following statement mean on page 5: “We further adapted AWP [Wu et al, 2020] so that the input data is never attacked.” Does this mean that there were no adversarial input robustness evaluations, or that AWP [Wu et al, 2020] was implemented by the authors in a way that did not harness adversarially crafted input examples during training with adversarial weight perturbation? If it is the latter, than the AWP approach may possibly be under-performing as well. Did the authors investigate that?

One recent concept that the authors should also discuss the similarities in their paper: In principle the idea is similar to training quantized neural networks with adversarial bit errors [Stutz et al., “Random and Adversarial Bit Error Robustness: Energy-Efficient and Secure DNN Accelerators”, arXiv preprint arXiv:2104.08323 (2021)], hence perturbing the weights during training for generalization to random weight perturbations at test time. Present study approaches this problem from the perspective of a network which operates with floating-point weights.

Authors use the abbreviation LSNN in the figures and tables, however this was never described in the paper. Instead the text used the abbreviation SRNNs. It is later discovered by the reader that LSNN abbreviation corresponds to the one by [Bellec et al. 2018], when one dives into the supplementary materials.


**Summary Of The Paper:**

Authors investigate the problem of network robustness in the presence of parameter variations. To that end, authors propose an adversarial parameter perturbation based robust training method. Proposed training approach iteratively performs adversarial attacks on the parameter space during training to regularize the model by penalizing parameter vulnerability. Experiments are performed on F-MNIST, ECG data and a speech command detection dataset, with both conventional CNNs and recurrent SNNs, where weight-space perturbations during training showed generalization to weight perturbations at inference time.

**Summary Of The Review:**

The paper has solid contributions by presenting an adversarial parameter perturbation based model regularization scheme that is successfully applied to recurrent SNNs. However there are important methodological comparisons/discussions missing with respect to similar methods that were recently proposed. My major concerns are listed in the above reviews, which concludes that I find this work currently to be below borderline.

---

> ### Author Response · Authors · 2021-11-18
> **Additional experiments including Resnet32, AMP, revised AWP**
>
> _So far the conventional CNN that the authors used only has 500k trainable parameters, and is not even demonstrated to operate on RGB images such as CIFAR-10. Did the authors investigate this with respect to the competitive methods?_
>
>
> - The reviewer correctly identifies that our algorithm induces significant training overhead that scales linearly with the number of steps used in the inner maximisation loop.
>     In order to address these concerns, we performed additional experiments on Cifar10 using Resnet32 and demonstrated the effectiveness of the algorithm when fewer steps are used (3 steps was the sweet spot, but one step still worked).
>     Additionally, we demonstrate that the algorithm also works on a pre-trained model, furthermore reducing the overall training overhead (see SM).
>
> - We would argue that the induced overhead is a price one is willing to pay in a scenario where a network is trained only once before being deployed to millions of devices.
>     This is however not the case when the proposed algorithm is thought of as a general method to increase generalisation, and it is for that reason why we demonstrate our algorithms using simulators that closely follow the noise models of actual neuromorphic hardware (see PCM model in SM).
>
> _[...] one needs to compare/discuss AMP with the proposed method (which uses a novel TRADES-like loss objective)_
>
> - We thank the reviewer for highlighting this important related work.
>     We have conducted additional experiments using AMP and show that AMP produces results that are inferior to those produced by our method (see updated Table 1). We believe this is one of the consequences of our design choice that lets us trade-off performance and robustness and the fact that the $\Delta \Theta$ found by the adversary is treated as a constant in the gradient computation. To ensure fairness, we conducted a parameter sweep of the attack size.
>
> _Does this mean that there were no adversarial input robustness evaluations, or that AWP [Wu et al, 2020] was implemented by the authors in a way that did not harness adversarially crafted input examples during training with adversarial weight perturbation? If it is the latter, than the AWP approach may possibly be under-performing as well. Did the authors investigate that?_
>
> - We thank the reviewer for highlighting this and the reviewer is correct in assuming that we omitted the adversarial attack in the input space.
>     In the initial setup we (falsely) assumed that this would only harm performance.
>     We have corrected this assumption and performed a sweep of the attack size for PGA in the input space.
>     We have found that this improved the performance for the Speech LSNN and have updated the results in Table 1 accordingly.
>     However, the corrected version still performs worse compared with other methods, including our proposed algorithm.
>
> _One recent concept that the authors should also discuss the similarities in their paper: In principle the idea is similar to training quantized neural networks with adversarial bit errors_
>
> - We thank the reviewer for presenting this interesting paper, and we added a section in "Related Work" to discuss this method.
>
> _It is later discovered by the reader that LSNN abbreviation corresponds to the one by [Bellec et al. 2018], when one dives into the supplementary materials._
>
> - We thank the reviewer for pointing out this flaw, which we addressed by substituting the abbreviation SRNN with LSNN in the paper.

---

> > ### Comment · Reviewer_BXMW · 2021-11-22
> > **Thanks to the authors for responses**
> >
> > Thanks to the authors for their responses. I have carefully read the revisions. Please find below my comments.
> >
> > Corrected AWP [Wu et al., 2020] implementations and added AMP [Zheng et al., 2021] experiments are performed in response to my comments. Similarly the authors nicely performed “larger-scale” experiments with ResNet-32 on CIFAR-10. One thing that is still in the air for me is that the authors did not show comparisons to the methods in Table I for these large-scale ResNet-32 models, except only comparing their method against “training with Gaussian noise on parameters”. Existing methods to adversarially perturb CNN parameters may still be equivalently (or better) successful for this problem, and this question seems not answered yet. Overall, I only decided to increase my rating to slightly above acceptance threshold given the extensive revisions.
> >
> > Minor comments:
> > - One should define what FP baseline stands for (i.e., floating point) before using it.
> > - Subfigures in Fig 3  should be annotated in titles/axis labels better. First row and second row of plots correspond to different methods (mentioned in the caption), but in the figure it should be made clear as well.
> > - The following sentence need a reference supporting the argument: “Currently, the method that has proven to yield the best performance on CiM hardware is training with noise on the network parameters…”.

---

> > > ### Author Response · Authors · 2021-11-24
> > > **Additional method comparisons**
> > >
> > > We thank the reviewer for their comments and suggestions. We will certainly address the minor comments in a revision before publication.
> > >
> > > Also, we are currently running additional experiments on the ResNet-32 architecture, to compare our method against AMP and AWP on CIFAR-10. These are the "second runner-up" methods from the existing experiments, after our method and after forward-noise. We have no reason to believe that AMP or AWP methods would perform differently or better on ResNet-32 compared with the other architectures, but we hope that this will address the reviewer's concern about the method comparisons.
> > >
> > > Unfortunately we do not yet have results from these experiments, but we will update the reviewer as soon as results are available. We will include these results also in a revision before publication.

---

> > > > ### Author Response · Authors · 2021-11-29
> > > > **Re: Thanks to the authors for responses**
> > > >
> > > > In the additional experiments that we performed on Cifar10 using the Resnet32 architecture, we compared our method to AWP and AMP. Our method outperformed AMP on all noise models. However when using the parameter mismatch noise model as described in our paper, AWP outperformed our model in the face of parameter mismatch.
> > > >
> > > > Under the PCM simulated drift experiment, the accuracy of AWP and our model were comparable at the time of device programming, and our method retained better accuracy than AWP in the face of simulated PCM drift.
> > > >
> > > > Given that our method was outperformed on one noise model, we further fine-tuned and re-ran AWP on the experiments already reported in the manuscript, and found that here our method still outperformed AWP.
> > > >
> > > > We conclude that for CNN-based architectures the advantage of our method over AWP depends on the noise model. More work is required to fully understand when AWP performs better. We will add this conclusion to the discussion in the final version of our paper and will update the results with the aforementioned experiments.

---

### Author Response · Authors · 2021-11-18
**Additional Experiments**

We would like to thank all the reviewers for their constructive and positive feedback.
In order to further extend the applicability and therefore the potential impact of our paper, we performed several new experiments during the review process.

Although these new experiments address some issues raised by individual reviewers, we chose to post a general comment because we believe that these experiments are of interest to all reviewers.

Besides CMOS based neuromorphic chips that mainly implement spiking neural networks (SNNs), in-memory computing based devices perform matrix-vector-multiplications (MVMs) using a crossbar comprising many re-programmable resistors.
These re-programmable devices mainly suffer from three types of noise: Read noise, write noise and drift.
While read- and write noise mostly introduce Gaussian perturbations, drift, as the name already suggests, causes the conductance of the devices to drift towards zero.

The effect of this can be alleviated by employing global drift compensation (GDC), a method that multiplies the output of the MVM by some calibration value that is obtained at the time of inference.

For these additional experiments, we have implemented a PyTorch based simulator, that takes all these noise sources into account, simulates analog to digital (ADC), as well as digital to analog (DAC) converters, and implements GDC.
The models used in the simulator are highly accurate and have proven to predict the performance on actual hardware quite well [1].

Our additional results can be summarized by the following bullet points:

- As the reviewers have pointed out, our method induces an additional training cost that scales linearly with the number of attack steps. We have now shown that our method works with much fewer number of attack steps (even one attack step works, but three was a sweet spot) and that one can just as effectively start from a pre-trained model that only needs to be fine-tuned for improved robustness.

- We are, to the best of our knowledge, the first to demonstrate a network that outperforms the floating point baseline (trained without noise and evaluated on a standard PC) in a precise simulator of PCM-based neuromorphic chips.

- We sweep the attack size and regularization magnitude (beta robustness) and show that fine hyperparameter tuning is not necessary in order to outperform the noise-injection-based approach.

- All of these results are demonstrated on Resnet32 trained on Cifar10, a well-known benchmark in the ML community.

We believe that these additional results greatly increase the applicability and therefore the strength of our paper.

[1] Collective structural relaxation in phase-change memory devices (Le Gallo et al.), _Advanced Electronic Materials_

[2] Accurate deep neural network inference using computational phase-change memory, (Joshi et al.), _Nature communications_

---

> ### Author Response · Authors · 2021-12-07
> **Update to additional experiments**
>
> In the additional experiments that we performed on Cifar10 using the Resnet32 architecture, we compared our method to AWP and AMP. Our method outperformed AMP on all noise models. However when using the parameter mismatch noise model as described in our paper, AWP outperformed our model in the face of parameter mismatch.
>
> Under the PCM simulated drift experiment, the accuracy of AWP and our model were comparable at the time of device programming, and our method retained better accuracy than AWP in the face of simulated PCM drift.
>
> Given that our method was outperformed on one noise model, we further fine-tuned and re-ran AWP on the experiments already reported in the manuscript, and found that here our method still outperformed AWP.
>
> We conclude that for CNN-based architectures the advantage of our method over AWP depends on the noise model. More work is required to fully understand when AWP performs better. We will add this conclusion to the discussion in the final version of our paper and will update the results with the aforementioned experiments.

---

### Decision · Program_Chairs · 2022-01-20

**Decision:**

Accept (Poster)

**Comment:**

The authors propose an adversarial training method to increase network robustness to parameter variations. The proposed approach performs adversarial attacks on network parameters during training. They demonstrate that their method flattens the loss landscape of the network. Experiments were performed on F-MNIST, ECG data, and speech command detection datasets using a conventional CNN and a recurrent spiking neural networks (SNNs).

The manuscript is well-written and the method is interesting.

One reviewer was somewhat concerned about the novelty of the work, but acknowledged that the application to recurrent SNNs was new.
The main initial criticism was the question of scalability of the method, as it was tested only on networks with a relatively small number of parameters.

In the revision, the authors addressed these issues. Their method was compared to related approaches, and experiments on CIFAR-10 with a ResNet32 were performed.
The reviewers acknowledged these larger-size experiments, but were not fully convinced as much larger models are typically used today.

Nevertheless, the reviewers acknowledged the improvements and ratings were increased, so all are voting for acceptance.